# Towards Impartial Multi-task Learning

**Liyang Liu**[1], **Yi Li**[2], **Zhanghui Kuang**[2], **Jing-Hao Xue**[3],
**Yimin Chen**[2], **Wenming Yang**[1],[*]**Qingmin Liao**[1], **Wayne Zhang**[2,4]
[1]Shenzhen International Graduate School/Department of
Electronic Engineering, Tsinghua University
[2]SenseTime Research
[3]Department of Statistical Science, University College London
[4]Qing Yuan Research Institute, Shanghai Jiao Tong University
{liu-ly14@mails., yang.wenming@sz., liaoqm@}tsinghua.edu.cn
{liyi, kuangzhanghui, chenyimin, wayne.zhang}@sensetime.com
jinghao.xue@ucl.ac.uk

## Abstract

Multi-task learning (MTL) has been widely used in representation learning. However, naïvely training all tasks simultaneously may lead to the partial training issue, where specific tasks are trained more adequately than others. In this paper, we propose to learn multiple tasks impartially. Specifically, for the *task-shared* parameters, we optimize the scaling factors via a closed-form solution, such that the aggregated gradient (sum of raw gradients weighted by the scaling factors) has equal projections onto individual tasks. For the *task-specific* parameters, we dynamically weigh the task losses so that all of them are kept at a comparable scale. Further, we find the above *gradient* balance and *loss* balance are complementary and thus propose a hybrid balance method to further improve the performance. Our impartial multi-task learning (IMTL) can be end-to-end trained without any heuristic hyper-parameter tuning, and is general to be applied on all kinds of losses without any distribution assumption. Moreover, our IMTL can converge to similar results even when the task losses are designed to have different scales, and thus it is scale-invariant. We extensively evaluate our IMTL on the standard MTL benchmarks including Cityscapes, NYUv2 and CelebA. It outperforms existing loss weighting methods under the same experimental settings.

## 1 Introduction

Recent deep networks in computer vision can match or even surpass human beings on some specific tasks separately. However, in reality multiple tasks (*e.g.*, semantic segmentation and depth estimation) must be solved simultaneously. Multi-task learning (MTL) (Caruana, 1997; Evgeniou & Pontil, 2004; Ruder, 2017; Zhang & Yang, 2017) aims at sharing the learned representation among tasks (Zamir et al., 2018) to make them benefit from each other and achieve better results and stronger robustness (Zamir et al., 2020). However, sharing the representation can lead to a partial learning issue: some specific tasks are learned well while others are overlooked, due to the different loss scales or gradient magnitudes of various tasks and the mutual competition among them. Several methods have been proposed to mitigate this issue either via *gradient balance* such as gradient magnitude normalization (Chen et al., 2018) and Pareto optimality (Sener & Koltun, 2018), or *loss balance* like homoscedastic uncertainty (Kendall et al., 2018). Gradient balance can evenly learn task-shared parameters while ignoring task-specific ones. Loss balance can prevent MTL from being biased in favor of tasks with large loss scales but cannot ensure the impartial learning of the shared parameters. In this work, we find that gradient balance and loss balance are complementary, and combining the two balances can further improve the results. To this end, we propose *impartial* MTL (IMTL) via simultaneously balancing gradients and losses across tasks.

For gradient balance, we propose IMTL-G(rad) to learn the scaling factors such that the aggregated gradient of task-shared parameters has equal projections onto the raw gradients of individual tasks

---

[*]Corresponding author

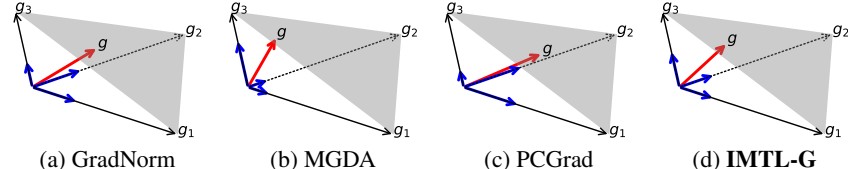

(a) GradNorm    (b) MGDA    (c) PCGrad    (d) **IMTL-G**

Figure 1: Comparison of gradient balance methods. In (a) to (d), $g_1$, $g_2$ and $g_3$ represent the gradient computed by the raw loss of each task, respectively. The gray surface represents the plane composed by these gradients. The red arrow denotes the aggregated gradient computed by the weighted sum loss, which is ultimately used to update the model parameters. The blue arrows show the projections of $g$ onto the raw gradients $\{g_t\}$. $g$ has the largest projection on $g_2$ (**nearest** to the mean direction), $g_3$ (**smallest** magnitude) and $g_2$ (**largest** magnitude) for GradNorm, MGDA and PCGrad, respectively, while the projections are **equal** on $\{g_t\}$ in our IMTL-G.

(see Fig. 1 (d)). We show that the scaling factor optimization problem is equivalent to finding the angle bisector of gradients from all tasks in geometry, and derive a closed-form solution to it. In contrast with previous gradient balance methods such as GradNorm (Chen et al., 2018), MGDA (Sener & Koltun, 2018) and PCGrad (Yu et al., 2020), which have learning biases in favor of tasks with gradients close to the average gradient direction, those with small gradient magnitudes, and those with large gradient magnitudes, respectively (see Fig. 1 (a), (b) and (c)), in our IMTL-G task-shared parameters can be updated without bias to any task.

For loss balance, we propose IMTL-L(oss) to automatically learn a loss weighting parameter for each task so that the weighted losses have comparable scales and the effect of different loss scales from various tasks can be canceled-out. Compared with uncertainty weighting (Kendall et al., 2018), which has biases towards regression tasks rather than classification tasks, our IMTL-L treats all tasks equivalently without any bias. Besides, we model the loss balance problem from the optimization perspective without any distribution assumption that is required by (Kendall et al., 2018). Therefore, ours is more general and can be used in any kinds of losses. Moreover, the loss weighting parameters and the network parameters can be jointly learned in an end-to-end fashion in IMTL-L.

Further, we find the above two balances are complementary and can be combined to improve the performance. Specifically, we apply IMTL-G on the task-shared parameters and IMTL-L on the task-specific parameters, leading to the hybrid balance method IMTL. Our IMTL is scale-invariant: the model can converge to similar results even when the same task is designed to have different loss scales, which is common in practice. For example, the scale of the cross-entropy loss in semantic segmentation may have different scales when using "average" or "sum" reduction over locations in the loss computation. We empirically validate that our IMTL is more robust against heavy loss scale changes than its competitors. Meanwhile, our IMTL only adds negligible computational overheads.

We extensively evaluate our proposed IMTL on standard benchmarks: Cityscapes, NYUv2 and CelebA, where the experimental results show that IMTL achieves superior performances under all settings. Besides, considering there lacks a fair and practical benchmark for comparing MTL methods, we unify the experimental settings such as image resolution, data augmentation, network structure, learning rate and optimizer option. We re-implement and compare with the representative MTL methods in a unified framework, which will be publicly available. Our contributions are:

- We propose a novel closed-form gradient balance method, which learns task-shared parameters without any task bias; and we develop a general learnable loss balance method, where no distribution assumption is required and the scale parameters can be jointly trained with the network parameters.

- We unveil that gradient balance and loss balance are complementary and accordingly propose a hybrid balance method to simultaneously balance gradients and losses.

- We validate that our proposed IMTL is loss scale-invariant and is more robust against loss scale changes compared with its competitors, and we give in-depth theoretical and experimental analyses on its connections and differences with previous methods.

- We extensively verify the effectiveness of our IMTL. For fair comparisons, a unified codebase will also be publicly available, where more practical settings are adopted and stronger performances are achieved compared with existing code-bases.

## 2 RELATED WORK

Recent advances in MTL mainly come from two aspects: network structure improvements and loss weighting developments. Network-structure methods based on soft parameter-sharing usually lead to high inference cost (review in Appendix A). Loss weighting methods find loss weights to be multiplied on the raw losses for model optimization. They employ a hard parameter-sharing paradigm (Ruder, 2017), where several light-weight task-specific heads are attached upon the heavy-weight task-agnostic backbone. There are also efforts that learn to group tasks and branch the network in the middle layers (Guo et al., 2020; Standley et al., 2020), which try to achieve better accuracy-efficiency trade-off and can be seen as semi-hard parameter-sharing. We believe task grouping and loss weighting are orthogonal and complementary directions to facilitate multi-task learning and can benefit from each other. In this work we focus on loss weighting methods which are the most economic as almost all of the computations are shared across tasks, leading to high inference speed. Task Prioritization (Guo et al., 2018) weights task losses by their difficulties to focus on the harder tasks during training. Uncertainty weighting (Kendall et al., 2018) models the loss weights as data-agnostic task-dependent homoscedastic uncertainty. Then loss weighting is derived from maximum likelihood estimation. GradNorm (Chen et al., 2018) learns the loss weights to enforce the norm of the scaled gradient for each task to be close. MGDA (Sener & Koltun, 2018) casts multi-task learning as multi-object optimization and finds the minimum-norm point in the convex hull composed by the gradients of multiple tasks. Pareto optimality is supposed to be achieved under mild conditions. GLS (Chennupati et al., 2019) instead uses the geometric mean of task-specific losses as the target loss, we will show it actually weights the loss by its reciprocal value. PCGrad (Yu et al., 2020) avoids interferences between tasks by projecting the gradient of one task onto the normal plane of the other. DSG (Lu et al., 2020) dynamically makes a task "stop or go" by its converging state, where a task is updated only once for a while if it is stopped. Although many loss weighting methods have been proposed, they are seldom open-sourced and rarely compared thoroughly under practical settings where strong performances are achieved, which motivates us to give an in-depth analysis and a fair comparison about them.

## 3 IMPARTIAL MULTI-TASK LEARNING

In MTL, we map a sample $x \in \mathbb{X}$ to its labels $\{y_t \in \mathbb{Y}_t\}_{t \in [1,T]}$ of all $T$ tasks through multiple task-specific mappings $\{f_t : \mathbb{X} \to \mathbb{Y}_t\}$. In most loss weighting methods, the hard parameter-sharing paradigm is employed, such that $f_t$ is parameterized by heavy-weight task-shared parameters $\theta$ and light-weight task-specific parameters $\theta_t$. All tasks take the same shared intermediate feature $z = f(x; \theta)$ as input, and the $t$-th task head outputs the prediction as $f_t(x) = f_t(z; \theta_t)$. We aim to find the scaling factors $\{\alpha_t\}$ for all $T$ task losses $\{L_t(f_t(x), y_t)\}$, so that the weighted sum loss $L = \sum_t \alpha_t L_t$ can be optimized to make all tasks perform well. This poses great challenges because: 1) losses may have distinguished forms such as cross-entropy loss and cosine similarity; 2) the dynamic ranges of losses may differ by orders of magnitude. In this work, we propose a hybrid solution for both the task-shared parameters $\theta$ and the task-specific parameters $\{\theta_t\}$, as Fig. 2.

### 3.1 GRADIENT BALANCE: IMTL-G

For task-shared parameters $\theta$, we can receive $T$ gradients $\{g_t = \nabla_\theta L_t\}$ via back-propagation from all of the $T$ raw losses $\{L_t\}$, and these gradients represent optimal update directions for individual tasks. As the parameters $\theta$ can only be updated with a single gradient, we should compute an aggregated gradient $g$ by the linear combination of $\{g_t\}$. It also implies to find the scaling factors $\{\alpha_t\}$ of raw losses $\{L_t\}$, since $g = \sum_t \alpha_t g_t = \nabla_\theta L = \nabla_\theta (\sum_t \alpha_t L_t)$. Motivated by the principle of balance among tasks, we propose to make the projections of $g$ onto $\{g_t\}$ to be equal, as Fig. 1 (d). In this way,

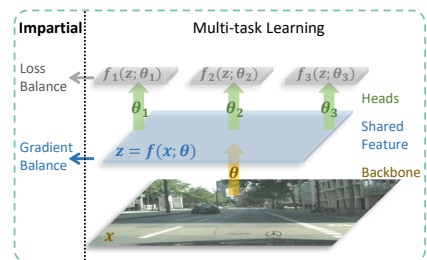

Figure 2: Overview of IMTL.

---

**Algorithm 1** Training by Impartial Multi-task Learning

---

**Input:** input sample $x$, task-specific labels $\{y_t\}$ and learning rate $\eta$
**Output:** task-shared/-specific parameters $\theta/\{\theta_t\}$, scale parameters $\{s_t\}$
 1: compute task-shared feature $z = f(x; \theta)$
 2: **for** $t = 1$ to $T$ **do**
 3:     compute task prediction by head network $f_t(x) = f_t^{\text{net}}(z; \theta_t)$
 4:     compute raw loss by loss function $L_t^{\text{raw}} = L_t^{\text{func}}(f_t(x), y_t)$
 5:     compute scaled loss $L_t = ba^{s_t} L_t^{\text{raw}} - s_t$ (default $a = e, b = 1$)          ▷ loss balance
 6:     compute gradient of shared feature $z$: $g_t = \nabla_z L_t$
 7:     compute unit-norm gradient $u_t = \frac{g_t}{\|g_t\|}$
 8: **end for**
 9: compute gradient differences $D^\top = [g_1^\top - g_2^\top, \cdots, g_1^\top - g_T^\top]$
10: compute unit-norm gradient differences $U^\top = [u_1^\top - u_2^\top, \cdots, u_1^\top - u_T^\top]$
11: compute scaling factors for tasks 2 to $T$: $\alpha_{2:T} = g_1 U^\top (DU^\top)^{-1}$          ▷ gradient balance
12: compute scaling factors for all tasks: $\alpha = [\ 1 - \mathbf{1}\alpha_{2:T}^\top, \quad \alpha_{2:T}\ ]$
13: update task-shared parameters $\theta = \theta - \eta\nabla_\theta(\sum_t \alpha_t L_t)$
14: **for** $t = 1$ to $T$ **do**
15:     update task-specific parameters $\theta_t = \theta_t - \eta\nabla_{\theta_t} L_t$
16:     update loss scale parameter $s_t = s_t - \eta\frac{\partial L_t}{\partial s_t}$
17: **end for**

---

we treat all tasks equally so that they progress in the same speed and none is left behind. Formally, let $\{u_t = g_t/\|g_t\|\}$ denote the unit-norm vector of $\{g_t\}$ which are row vectors, then we have:

$$gu_1^\top = gu_t^\top \Leftrightarrow g(u_1 - u_t)^\top = 0, \ \forall\, 2 \leqslant t \leqslant T. \tag{1}$$

The above problem is under-determined, but we can obtain the closed-form results of $\{\alpha_t\}$ by constraining $\sum_t \alpha_t = 1$. Assume $\alpha = [\alpha_2, \cdots, \alpha_T]$, $U^\top = [u_1^\top - u_2^\top, \cdots, u_1^\top - u_T^\top]$, $D^\top = [g_1^\top - g_2^\top, \cdots, g_1^\top - g_T^\top]$ and $\mathbf{1} = [1, \cdots, 1]$, from Eq. (1) we can obtain:

$$\alpha = g_1 U^\top (DU^\top)^{-1}. \qquad \text{(IMTL-G)} \tag{2}$$

The detailed derivation is in Appendix B.1. After obtaining $\alpha$, the scaling factor of the first task can be computed by $\alpha_1 = 1 - \mathbf{1}\alpha^\top$ since $\sum_t \alpha_t = 1$. The optimized $\{\alpha_t\}$ are used to compute $L = \sum_t \alpha_t L_t$, which is ultimately minimized by SGD to update the model. By now, back-propagation needs to be executed $T$ times to obtain the gradient of each task loss with respect to the heavy-weight task-shared parameters $\theta$, which is time-consuming and non-scalable. We replace the parameter-level gradients $\{g_t = \nabla_\theta L_t\}$ with feature-level gradients $\{\nabla_z L_t\}$ to compute $\{\alpha_t\}$. This implies to achieve gradient balance with respect to the last shared feature $z$ as a surrogate of task-shared parameters $\theta$, since it is possible for the network to back-propagate this balance all the way through the task-shared backbone starting from $z$. This relaxation allows us to do back propagation through the backbone only once after obtaining $\{\alpha_t\}$, and thus the training time can be dramatically reduced.

## 3.2 Loss Balance: IMTL-L

For the task-specific parameters $\{\theta_t\}$, we cannot employ IMTL-G described above, because $\nabla_{\theta_t} L_\tau = \mathbf{0}, \ \forall t \neq \tau$, and thus only the gradient of the corresponding task $\nabla_{\theta_t} L_t$ can be obtained for each $\theta_t$. Instead we propose to balance the losses among tasks by forcing the scaled losses $\{\alpha_t L_t\}$ to be constant for all tasks, without loss of generality, we take the constant as 1. Then the most direct idea is to compute the scaling factors as $\{\alpha_t = 1/L_t\}$, but they are sensitive to outlier samples and manifest severe oscillations, so we further propose to *learn* to scale losses via gradient descent and thus stronger stability can be achieved. Suppose the positive losses $\{L_t > 0\}$ are to be balanced, we first introduce a mapping function $h : \mathbb{R} \to \mathbb{R}^+$ to transform the arbitrarily-ranged learnable scale parameters $\{s_t\}$ to positive scaling factors $\{h(s_t) > 0\}$, hereafter we abandon the subscript $t$ for brevity. Then we should construct an appropriate scaled loss $g(s)$ so that *both* network parameters $\theta$ and scale parameter $s$ can be optimized by *minimizing* $g(s)$. On one hand, we balance different

tasks by encouraging the scaled losses $h(s) L(\boldsymbol{\theta})$ to be 1 for all tasks, so the optimality $s^\star$ of $s$ is achieved when $h(s) L(\boldsymbol{\theta}) = 1$, or equivalently:

$$f(s) \equiv h(s) L(\boldsymbol{\theta}) - 1 = 0, \text{ if } s = s^\star. \tag{3}$$

One may expect to minimize $|f(s)| = |h(s) L(\boldsymbol{\theta}) - 1|$ to find $s^\star$, however when $h(s) L(\boldsymbol{\theta}) < 1$, the gradient with respect to $\boldsymbol{\theta}$, $\nabla_{\boldsymbol{\theta}} |f(s)| = -h(s) \nabla_{\boldsymbol{\theta}} L(\boldsymbol{\theta})$, is in the opposite direction. On the other hand, assume our scaled loss $g(s)$ is a differentiable convex function with respect to $s$, then its minimum is achieved if and only if $s = s^\star$, where the derivative of $g(s)$ is zero:

$$g'(s) = 0, \text{ if } s = s^\star. \tag{4}$$

From Eq. (3) and (4) we find that the values of $f(s)$ and $g'(s)$ are both 0 when $s = s^\star$, we can then regard $f(s)$ as the derivative of $g(s)$, which is our target scaled loss and used to optimize both the network parameters $\boldsymbol{\theta}$ and loss scale parameter $s$, then we have:

$$g'(s) = f(s) \Leftrightarrow g(s) = \int f(s)\, \mathrm{d}s = L(\boldsymbol{\theta}) \int h(s)\, \mathrm{d}s - s. \tag{5}$$

From Eq. (3) and (5), we notice that both $h(s)$ and $\int h(s)\, \mathrm{d}s$ denote loss scales, so we have $\int h(s)\, \mathrm{d}s = Ch(s)$, where $C > 0$ is a constant. According to ordinary differential equation, $\int h(s)\, \mathrm{d}s$ must be the exponential function: $\int h(s)\, \mathrm{d}s = ba^s$ with $a > 1, b > 0$ (see Appendix B.2). We then have $g''(s) = ka^s$, $k > 0$, which is always positive and verifies our assumption about the convexity of $g(s)$. Also note that the gradient of $g(s)$ with respect to $\boldsymbol{\theta}$, $\nabla_{\boldsymbol{\theta}} g(s) = \int h(s)\, \mathrm{d}s \nabla_{\boldsymbol{\theta}} L(\boldsymbol{\theta}) = ba^s \nabla_{\boldsymbol{\theta}} L(\boldsymbol{\theta})$, is in the appropriate direction since $ba^s > 0$. As an instantiation, we set $\int h(s)\, \mathrm{d}s = e^s$ $(a = e, b = 1)$, then

$$g(s) = e^s L(\boldsymbol{\theta}) - s, \qquad \text{(IMTL-L)}. \tag{6}$$

From Eq. (6) we find that the raw loss is scaled by $e^s$, and $-s$ acts as a regularization to avoid the trivial solution $s = -\infty$ while minimizing the scaled loss $g(s)$. As for implementation, the task losses $\{L_t\}$ are scaled by $\{e^{s_t}\}$, and the scaled losses $\{e^{s_t} L - s_t\}$ are used to update both the network parameters $\boldsymbol{\theta}$, $\{\boldsymbol{\theta}_t\}$ and the scale parameters $\{s_t\}$.

### 3.3 Hybrid Balance: IMTL

We have introduced IMTL-G/IMTL-L to achieve gradient/loss balance, and both of them produce scaling factors to be applied on the raw losses. They can be used solely, but we find them complementary and able to be combined to improve the performance. In IMTL-G, even if the raw losses are multiplied by arbitrary (maybe different among tasks) positive factors, the direction of the aggregated gradient $\boldsymbol{g}$ stays unchanged. Because by definition $\boldsymbol{g} = \sum_t \alpha_t \boldsymbol{g}_t$ is the angular bisector of the gradients $\{\boldsymbol{g}_t\}$, and positive scaling will not change the directions of $\{\boldsymbol{g}_t\}$ and thus that of $\boldsymbol{g}$ (proof in Theorem 2). So we can also obtain the scale factors $\{\alpha_t\}$ in IMTL-G with the losses that have been scaled by $\{s_t\}$ from IMTL-L. IMTL-G and IMTL-L are combined as: 1) the task-specific parameters $\{\boldsymbol{\theta}_t\}$ and scale parameters $\{s_t\}$ are updated by scaled losses $\{e^{s_t} L_t - s_t\}$; 2) the task-shared parameters $\boldsymbol{\theta}$ are updated by $\sum_t \alpha_t (e^{s_t} L_t)$ which is the weighted average of $\{e^{s_t} L_t\}$, with the weights $\{\alpha_t\}$ computed by $\{\nabla_{\boldsymbol{z}} (e^{s_t} L_t)\}$ using IMTL-G. Note that the regularization terms $\{-s_t\}$ in Eq. (6) are constants with respect to $\boldsymbol{\theta}$ and $\boldsymbol{z}$, and thus can be ignored when computing gradients and updating parameters in IMTL-G. In this way, we achieve both gradient balance for task-shared parameters and loss balance for task-specific parameters, leading to our full IMTL as illustrated in Alg. 1.

## 4 Discussion

We draw connections between our method and previous state-of-the-arts [1] in Fig. 3. We will show that previous methods can all be categorized as gradient or loss balance, and thus each of them can be seen as a specification of our method. However, all of them have some intrinsic biases or short-comings leading to inferior performances, which we try to overcome.

---

[1] Our analysis of PCGrad (Yu et al., 2020) can be found in Appendix C.3.

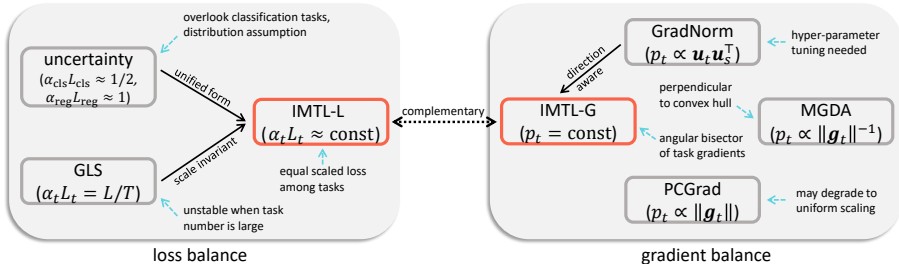

Figure 3: Relationship between our IMTL and previous methods. The blue dashed arrow indicates the characteristic of each method. In the *loss balance* methods, we annotate the scaled loss in the bracket. $L_{\text{cls}}$, $L_{\text{reg}}$ and $L_t$ are the raw loss of classification, regression and individual task, respectively. $\alpha_{\text{cls}}$, $\alpha_{\text{reg}}$ and $\alpha_t$ is the corresponding loss scale. $L$ is the geometric mean loss and $T$ is the task number. In the *gradient balance* methods, we annotate the projections of the aggregated gradient $\boldsymbol{g} = \sum_t \alpha_t \boldsymbol{g}_t$ onto the raw gradient $\boldsymbol{g}_t$ of the $t$-th task in the bracket. $\boldsymbol{u}_t = \boldsymbol{g}_t / \|\boldsymbol{g}_t\|$ is the unit-norm vector, $p_t = \boldsymbol{g}\boldsymbol{u}_t^\top$ is the projection of $\boldsymbol{g}$ onto $\boldsymbol{g}_t$ and $\boldsymbol{u}_s = \sum_t \boldsymbol{u}_t$ is the mean direction.

**GradNorm** (Chen et al., 2018) balances tasks by making the norm of the scaled gradient for each task to be approximately equal. It also introduces the inverse training rate and a hyper-parameter $\gamma$ to control the strength of approaching the mean gradient norm, such that tasks which learn slower can receive larger gradient magnitudes. However, it does not take into account the relationship of the gradient directions. We show that when the angle between the gradients of each pair of tasks is identical, our IMTL-G leads to the equivalent solution as GradNorm.

**Theorem 1.** *If the angle between any pair of $\boldsymbol{u}_t, \boldsymbol{u}_\tau$ stays constant: $\boldsymbol{u}_t\boldsymbol{u}_\tau^\top = C_1$, $\forall t \neq \tau$ with $C_1 < 1$, then our IMTL-G leads to the same solution as that of GradNorm: $\boldsymbol{g}\boldsymbol{u}_t^\top = C_2 \Leftrightarrow n_t \equiv \|\alpha_t \boldsymbol{g}_t\| = \alpha_t \|\boldsymbol{g}_t\| = C_3$. In the above $\boldsymbol{u}_t = \boldsymbol{g}_t / \|\boldsymbol{g}_t\|$, $C_1$, $C_2$ and $C_3$ are constants.*

Proof in Appendix C.1. In GradNorm, if without the above constant-angle condition $\boldsymbol{u}_t\boldsymbol{u}_\tau^\top = C_1$, the projection of the aggregated gradient $\boldsymbol{g}$ onto task-specific gradient, $\boldsymbol{g}\boldsymbol{u}_t^\top = \left(\sum_\tau C_3 \boldsymbol{u}_\tau\right)\boldsymbol{u}_t^\top = C_3\left(\sum_\tau \boldsymbol{u}_\tau\right)\boldsymbol{u}_t^\top$, is proportional to $\left(\sum_\tau \boldsymbol{u}_\tau\right)\boldsymbol{u}_t^\top$. It tends to optimize the "majority tasks" whose gradient directions are closer to the mean direction $\sum_t \boldsymbol{u}_t$, resulting in undesired task bias.

**MGDA** (Sener & Koltun, 2018) finds the weighted average gradient $\boldsymbol{g} = \sum_t \alpha_t \boldsymbol{g}_t$ with minimum norm in the convex hull composed by $\{\boldsymbol{g}_t\}$, so that $\sum_t \alpha_t = 1$ and $\alpha_t \geqslant 0$, $\forall t$. It adopts an iterative method based on Frank-Wolfe algorithm to solve the multi-objective optimization problem. We note the minimum-norm point has a closed-form representation if without the constraints $\{\alpha_t \geqslant 0\}$. In this case, we try to minimize $\boldsymbol{g}\boldsymbol{g}^\top = \left(\sum_t \alpha_t \boldsymbol{g}_t\right)\left(\sum_\tau \alpha_\tau \boldsymbol{g}_\tau\right)^\top$ such that $\sum_t \alpha_t = 1$. It implies $\boldsymbol{g}$ is perpendicular to the hyper-plane composed by $\{\boldsymbol{g}_t\}$ as illustrated in Fig 1 (b), and thus we have:

$$\boldsymbol{g} \perp (\boldsymbol{g}_1 - \boldsymbol{g}_t) \Leftrightarrow \boldsymbol{g}(\boldsymbol{g}_1 - \boldsymbol{g}_t)^\top = 0, \ \forall\, 2 \leqslant t \leqslant T, \tag{7}$$

and can obtain $\boldsymbol{\alpha} = \boldsymbol{g}_1 \boldsymbol{D}^\top \left(\boldsymbol{D}\boldsymbol{D}^\top\right)^{-1}$ (see Appendix C.2). From Eq. (7), we note that the aggregated gradient satisfies: $\boldsymbol{g}\boldsymbol{g}_t^\top = C$. Then the projection of $\boldsymbol{g}$ onto $\boldsymbol{g}_t$, $\boldsymbol{g}\boldsymbol{u}_t^\top = C / \|\boldsymbol{g}_t\|$, is inversely proportional to the norm of $\boldsymbol{g}_t$. So it focuses on tasks with smaller gradient magnitudes, which breaks the task balance. Even with $\{\alpha_t \geqslant 0\}$, the problem still exists (see Appendix C.2) in the original MGDA method. Through experiments, we note that finding the minimum-norm point without the constraints $\{\alpha_t \geqslant 0\}$ leads to similar performance as MGDA with the constraints $\{\alpha_t \geqslant 0\}$. In our IMTL-G, although we do not constrain $\{\alpha_t \geqslant 0\}$, its loss weighting scales are always positive during the training procedure as shown in Fig. 4.

**Uncertainty weighting** (Kendall et al., 2018) regards the task uncertainty as loss weight. For regression, it can derive $L_1$ loss from Laplace distribution: $-\log p\left(y \mid f\left(\boldsymbol{x}\right)\right) = |y - f\left(\boldsymbol{x}\right)| / b + \log b$, where $\boldsymbol{x}$ is the data sample, $y$ is the ground-truth label, $f$ denotes the prediction model and $b$ is the diversity of Laplace distribution. $L_2$ loss can be found in Appendix C.4. For classification, it takes the cross-entropy loss as a scaled categorical distribution and introduces the following approximation:

$$-\log p\left(y \mid f\left(\boldsymbol{x}\right)\right) = -\log\left[\text{softmax}_y\left(\frac{f\left(\boldsymbol{x}\right)}{\sigma^2}\right)\right] \approx -\frac{1}{\sigma^2}\log\left[\text{softmax}_y\left(f\left(\boldsymbol{x}\right)\right)\right] + \log\sigma, \tag{8}$$

in which $\text{softmax}_y(\cdot)$ stands for taking the $y$-th entry after the $\text{softmax}(\cdot)$ operator. MTL corresponds to maximizing the joint likelihood of multiple targets, then the derivations yield the scaling factor $b/\sigma$ for the regression/classification loss. (Kendall et al., 2018) learn $b$ and $\sigma$ as model parameters which are updated by stochastic gradient descent. However, it is applicable only if we can find appropriate correspondence between the loss and the distribution. It is difficult to be used for losses such as cosine similarity, and it is impossible to traverse all kinds of losses to obtain a unified form for them. Moreover, it sacrifices classification tasks. From Eq. (8) we can find that the scaled cross-entropy loss is approximated as $L = e^{2s}L_{\text{cls}} - s$ if we set $s = -\log \sigma$. By taking the derivative we have $\partial L/\partial s = 2e^{2s}L_{\text{cls}} - 1$. Then $s$ is optimized to make the scaled loss $e^{2s}L_{\text{cls}}$ to be close to $1/2$. However, the scaled $L_1$ loss is approximated as $L = e^{s}L_{\text{reg}} - s$ if we set $s = -\log b$, and taking the derivative we have $\partial L/\partial s = e^{s}L_{\text{reg}} - 1$. So $s$ is optimized to make the scaled $L_1$ loss to achieve $1$, which is twice of the classification loss, and thus the classification task is overlooked.

We would like to remark the differences between our IMTL-L and uncertainty weighting (Kendall et al., 2018). **Firstly**, our derivation is motivated by the fairness among tasks, which intrinsically differs from uncertainty weighting which is based on task uncertainty considering each task independently. **Secondly**, IMTL-L learns to balance among tasks without any biases, while uncertainty weighting may sacrifice classification tasks to favor regression tasks as derived above. **Thirdly**, IMTL-L does not depend on any distribution assumptions and thus can be generally applied to various losses including cosine similarity, which uncertainty weighting may have difficulty with. As far as we know, there is no appropriate correspondence between cosine similarity and specific distributions. **Lastly**, uncertainty weighting needs to deal with different losses case by case, it also introduces approximations in order to derive scaling factors for certain losses (such as cross-entropy loss) which may not be optimal, but our IMTL-L has a unified form for all kinds of losses.

**GLS** (Chennupati et al., 2019) calculates the target loss as the geometric mean: $L = \left(\prod_t L_t\right)^{\frac{1}{T}}$, then the gradient of $L$ with respect to the model parameters $\boldsymbol{\theta}$ can be obtained as Appendix C.5, which can be regarded as to weigh the loss with its reciprocal value. However, as the gradient depends on the value of $L$, so it is not scale-invariant to the loss scale changes. Moreover, we find it to be unstable when the number of tasks is large because of the geometric mean computation.

## 5 EXPERIMENTS

In previous methods, various experimental settings have been adopted but there are no extensive comparisons. As one contribution of our work, we re-implement representative methods and present fair comparisons among them under the unified code-base, where more practical settings are adopted and stronger performances are achieved compared with existing code-bases. The implementations exactly follow the original papers and open-sourced code to ensure the correctness. We run experiments on the Cityscapes (Cordts et al., 2016), NYUv2 (Silberman et al., 2012) and CelebA (Liu et al., 2015) dataset to extensively analyze different methods. Details can be found in Appendix D.

**Results on Cityscapes.** From Tab. 1 we can obtain several informative conclusions. The uniform scaling baseline, which naïvely adds all losses, tends to optimize tasks with larger losses and gradient magnitudes, resulting in severe task bias. Uncertainty weighting (Kendall et al., 2018) sacrifices classification tasks to aid regression ones, leading to significantly worse results on semantic segmentation compared with our IMTL-L. GradNorm (Chen et al., 2018) is very sensitive to the choice of the hyper-parameter $\gamma$ controlling the strength of equal gradient magnitudes, where the default $\gamma = 1.5$ works well on NYUv2 but performs badly on Cityscapes. We find its best option is $\gamma = 0$ which makes the scaled gradient norm to be exactly equal. MGDA (Sener & Koltun, 2018) focuses on tasks with smaller gradient magnitudes. So the performance of semantic segmentation is good but the other two tasks have difficulty in converging. In addition, we find our proposed closed-form variant without the hard constraints $\{\alpha_t \geqslant 0\}$ achieves similar results as the original iterative method. Through the experiments we notice the closed-form solution almost always yields $\{\alpha_t \geqslant 0\}$. As for PCGrad (Yu et al., 2020), it yields slightly better performance than uniform scaling because its conflict projection will have no effect when the angles between the gradients are equal or less than $\pi/2$. In contrast, our IMTL method, in terms of both gradient balance and loss balance, yields competitive performance and achieves the best balance among tasks. Moreover, we verify that the two balances are complementary and can be combined to further improve the performance, with the visualizations in Appendix E. Surprisingly, we find our IMTL can beat the single-task baseline where

Table 1: Comparison between IMTL and previous methods on Cityscapes, **sem**antic segmentation, **ins**tance segmentation and **disp**arity/depth estimation are considered. The first group of columns shows the regular results of different methods. The second group shows the results by manually multiply the semantic segmentation loss with 10 before applying these methods. The subscript numbers show the absolute change after scaling the loss to demonstrate the robustness of various methods. The arrows indicate the values are the higher the better ($\uparrow$) or the lower the better ($\downarrow$). The best and runner up results for each task are bold and underlined, respectively.

| method | sem. mIoU$\uparrow$ | ins. $L_1 \downarrow$ | disp. $L_1 \downarrow$ | sem. mIoU$\uparrow_{|\Delta|\downarrow}$ | ins. $L_1 \downarrow_{|\Delta|\downarrow}$ | disp. $L_1 \downarrow_{|\Delta|\downarrow}$ | time s/iter$\downarrow$ |
|---|---|---|---|---|---|---|---|
| baselines | | | | | | | |
| single-task | 76.67 | 21.61 | 4.182 | - | - | - | - |
| uniform scaling | 58.99 | 18.13 | 3.512 | - | - | - | 1.201 |
| *loss balance* | | | | | | | |
| uncertainty (Kendall et al., 2018) | 74.91 | $\underline{16.43}$ | **2.895** | $74.00_{0.91}$ | $\underline{16.77}_{0.34}$ | **2.930**$_{0.035}$ | 1.204 |
| GLS (Chennupati et al., 2019) | 75.65 | 17.18 | 2.953 | $66.22_{9.43}$ | $21.09_{3.91}$ | $3.358_{0.405}$ | 1.202 |
| **IMTL-L** | 76.89 | 16.69 | 2.944 | $75.55_{1.34}$ | $17.49_{0.80}$ | $2.972_{0.028}$ | 1.202 |
| gradient *balance* | | | | | | | |
| GradNorm ($\gamma = 0$) | 76.27 | 17.99 | 3.195 | $72.96_{3.31}$ | $19.36_{1.37}$ | $3.216_{0.021}$ | 1.741 |
| GradNorm (Chen et al., 2018) | 52.17 | 19.88 | 4.098 | $54.23_{2.06}$ | $20.53_{0.65}$ | $4.108_{0.010}$ | 1.742 |
| MGDA (w/o $\{\alpha_t \geqslant 0\}$) | $\underline{76.95}$ | 53.19 | 6.296 | $\underline{76.36}_{0.59}$ | $29.06_{24.13}$ | $3.377_{2.919}$ | 1.777 |
| MGDA (Sener & Koltun, 2018) | 76.56 | 53.14 | 6.644 | $72.35_{4.21}$ | $29.38_{23.76}$ | $3.336_{3.308}$ | 1.732 |
| PCGrad (Yu et al., 2020) | 60.50 | 17.99 | 3.450 | $66.33_{5.83}$ | $17.99_{0.00}$ | $3.386_{0.064}$ | 2.087 |
| IMTL-G (exact) | 76.13 | 17.46 | 2.979 | - | - | - | 2.769 |
| **IMTL-G** | 76.52 | 16.61 | 2.997 | $76.06_{0.46}$ | $17.52_{0.91}$ | $3.020_{0.023}$ | 1.776 |
| hybrid *balance* | | | | | | | |
| **IMTL** | **77.00** | **15.96** | $\underline{2.905}$ | $76.56_{0.44}$ | **15.85**$_{0.11}$ | $\underline{2.938}_{0.033}$ | 1.795 |

each task is trained with a separate model. Training multiple tasks simultaneously can learn a better representation from multiple levels of semantics, which can in turn improve individual tasks.

In addition, we present the real-world training time of each iteration for different methods in Tab. 1. As shown, loss balance methods are the most efficient, and our gradient balance method IMTL-G adds acceptable computational overhead, similar to that of GradNorm (Chen et al., 2018) and MGDA (Sener & Koltun, 2018). It benefits from computing gradients with respect to the shared feature maps instead of the shared model parameters (the row of "IMTL-G (exact)"), which brings similar performances but adds significant complexity due to multiple ($T$) backward passes through the shared parameters. Our IMTL-G only needs to do backward computation on the shared parameters once after obtaining the loss weights via Eq. (2), in which the computation overhead mainly comes from the matrix multiplication rather than the matrix inverse, since the inversed matrix $\boldsymbol{DU}^\top \in \mathbb{R}^{(T-1)\times(T-1)}$ is small compared with dimension of the shared feature $\boldsymbol{z}$.

As we outperform MGDA (Sener & Koltun, 2018) and PCGrad (Yu et al., 2020) significantly in terms of the objective metrics shown in Tab. 1, we further compare the qualitative results of our hybrid balance IMTL with the loss balance method uncertainty weighting (Kendall et al., 2018) and the gradient balance method GradNorm (Chen et al., 2018) considering their strong performances (see Fig. 6). For depth estimation we only show predictions at the pixels where ground truth (GT) labels exist to compare with GT, which is different from Fig. 7 where depth predictions are shown for all pixels. Consistent with results in Tab. 1, our IMTL shows visually noticeable improvements especially for the semantic and instance segmentation tasks. It is worth noting that we conduct experiments under strong baselines and practical settings which are seldom explored before, in this case changing the backbone in PSPNet (Zhao et al., 2017) from ResNet-50 to ResNet-101 can only improve mIoU of the semantic segmentation task around $0.5\%$ according to the public code base[2].

**Scale invariance.** We are also interested in the scale invariance, which means how the results change with the loss scale. For example, in semantic segmentation, the loss scale is different if we replace the reduction method "mean" (averaged over all locations) with "sum" (summed over all locations) in the cross-entropy loss computation, or the number of the interested classes increases. The scale invariance is beneficial for model robustness. So to simulate this effect, we manually multiply the semantic segmentation loss by 10 and apply the same methods to see how the performances are affected. In the last three columns of Tab. 1 we report the absolute changes resulting from the

---

[2]https://github.com/open-mmlab/mmsegmentation/tree/master/configs/pspnet

Table 2: Experimental results on the NYUv2 and CelebA datasets, **sem**antic segmentation, surface **norm**al estimation, **depth** estimation and multi-class **class**ification are considered. Arrows indicate the values are the higher the better (↑) or the lower the better (↓). The best and runner up results in each column are bold and underlined, respectively.

| method | NYUv2 | | | CelebA |
|---|---|---|---|---|
| | sem. | norm. | depth | class. |
| | mIoU↑ | cos↑ | $L_1 \downarrow$ | acc. ↑ |
| *baselines* | | | | |
| single-task | 56.82 | 0.8827 | 0.5097 | - |
| uniform scaling | 57.40 | 0.8684 | 0.4248 | 90.01 |
| *loss balance* | | | | |
| uncertainty (Kendall et al., 2018) | 57.20 | - | 0.4400 | 90.34 |
| GLS (Chennupati et al., 2019) | 57.84 | 0.8762 | 0.4243 | - |
| **IMTL-L** | 58.36 | 0.8864 | **0.4173** | 90.54 |
| *gradient balance* | | | | |
| GradNorm ($\gamma = 0$) | 55.96 | 0.8818 | 0.4317 | 90.91 |
| GradNorm (Chen et al., 2018) | 56.92 | 0.8787 | 0.4285 | 89.92 |
| MGDA (w/o $\{\alpha_t \geqslant 0\}$) | 49.43 | 0.8877 | 0.4839 | 89.68 |
| MGDA (Sener & Koltun, 2018) | 49.44 | 0.8875 | 0.4759 | 90.04 |
| PCGrad (Yu et al., 2020) | 57.48 | 0.8696 | 0.4253 | 89.99 |
| **IMTL-G** | 57.00 | 0.8785 | 0.4226 | 91.03 |
| *hybrid balance* | | | | |
| **IMTL** | **58.85** | **0.8888** | 0.4215 | **91.12** |

multiplier. Our IMTL achieves the smallest performance fluctuations and thus the best invariance, while other methods are more or less affected by the loss scale change.

**Results on NYUv2.** In Tab. 2 we find similar patterns as on Cityscapes, but NYUv2 is a rather small dataset, so uniform scaling can also obtain reasonable results. Note that uncertainty weighting (Kendall et al., 2018) cannot be directly used to estimate the normal surface when the cosine similarity is used as the loss, since no appropriate distribution can be found to correspond to cosine similarity. In this case, surface normal estimation owns the smallest gradient magnitude, so MGDA (Sener & Koltun, 2018) learns it best but it performs not so well for the rest two tasks. Again, our IMTL performs best taking advantage of the complementary gradient and loss balances.

**Results on CelebA.** To compare different methods in the many-task setting, in Tab. 2 we also conduct the multi-label classification experiments on the CelebA (Liu et al., 2015) dataset. The mean accuracy of 40 tasks is used as the final metric. Our IMTL outperforms its competitors in the scenario where the task number is large, showing its superiority. Note that in this setting, GLS (Chennupati et al., 2019) has difficulty in converging and no reasonable results can be obtained.

## 6 CONCLUSION

We propose an impartial multi-task learning method integrating gradient balance and loss balance, which are applied on task-shared and task-specific parameters, respectively. Through our in-depth analysis, we have theoretically compared our method with previous state-of-the-arts. We have also showed that those state-of-the-arts can all be categorized as gradient or loss balance, but lead to specific bias among tasks. Through extensive experiments we verify our analysis and demonstrate the effectiveness of our method. Besides, for fair comparisons, we contribute a unified code-base, which adopts more practical settings and delivers stronger performances compared with existing code-bases, and it will be publicly available for future research.

ACKNOWLEDGEMENTS

This work was supported by the Natural Science Foundation of Guangdong Province (No. 2020A1515010711), the Special Foundation for the Development of Strategic Emerging Industries of Shenzhen (No. JCYJ20200109143010272), and the Innovation and Technology Commission of the Hong Kong Special Administrative Region, China (Enterprise Support Scheme under the Innovation and Technology Fund B/E030/18).

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

## A    RELATED WORK OF NETWORK STRUCTURE

Cross-stitch Networks (Misra et al., 2016) learn coefficients to linearly combine activations from multiple tasks to construct better task-specific representations. To break the limitation of channel-wise cross-task feature fusion only, NDDR-CNN (Gao et al., 2019) proposes the layer-wise cross-channel feature aggregation as $1 \times 1$ convolutions on the concatenated feature maps from multiple tasks. More generally, MTL-NAS (Gao et al., 2020) introduces cross-layer connections among tasks to fully exploit the feature sharing from both low and high layers, extending the idea in Sluice Networks (Ruder et al., 2019) by leveraging neural architecture search (Zoph & Le, 2017). The parameters of these methods increase linearly with the number of tasks. To improve the model compactness, Residual Adapters (Rebuffi et al., 2017) introduce a small amount of task-specific parameters for each layer and convolve them with the task-agnostic representations to form the task-related ones. MTAN (Liu et al., 2019) generates data-dependent attention tensors by task-specific parameters to attend to the task-shared features. Single-tasking (Maninis et al., 2019) instead applies squeeze-and-excitation (Hu et al., 2018) module to generate attentive vectors for each task. In Task Routing (Strezoski et al., 2019), the attentive vectors are randomly sampled before training and are fixed for each image. Piggyback (Mallya et al., 2018) opts to mask parameter weights in place of activation maps, dealing with task-sharing from another point-of-view. The above methods can share parameters among tasks to a large extent, however, they are not memory-efficient because each task still needs to compute all of its own intermediate feature maps, which also leads to inferior inference speed compared with loss weighting methods.

## B    DETAILED DERIVATION

### B.1    GRADIENT BALANCE: IMTL-G

Here we give the detailed derivation of the closed-form solution of our IMTL-G, we also demonstrate the scale-invariance property of our IMTL-G, which is invariant to the scale changes of losses.

**Solution.** As we want to achieve:

$$\boldsymbol{g}\boldsymbol{u}_1^\top = \boldsymbol{g}\boldsymbol{u}_t^\top \Leftrightarrow \boldsymbol{g}\left(\boldsymbol{u}_1 - \boldsymbol{u}_t\right)^\top = 0, \ \forall \, 2 \leqslant t \leqslant T, \tag{9}$$

where $\boldsymbol{u}_t = \boldsymbol{g}_t / \|\boldsymbol{g}_t\|$, recall that we have $\boldsymbol{g} = \sum_t \alpha_t \boldsymbol{g}_t$ and $\sum_t \alpha_t = 1$, if we set $\boldsymbol{\alpha} = [\alpha_2, \cdots, \alpha_T]$ and $\boldsymbol{G}^\top = [\boldsymbol{g}_2^\top, \cdots, \boldsymbol{g}_T^\top]$, then $\alpha_1 = 1 - \mathbf{1}\boldsymbol{\alpha}^\top$ and Eq. (9) can be expanded as:

$$\left(\sum_t \alpha_t \boldsymbol{g}_t\right) \left[\boldsymbol{u}_1^\top - \boldsymbol{u}_2^\top, \cdots, \boldsymbol{u}_1^\top - \boldsymbol{u}_T^\top\right] = \mathbf{0} \Leftrightarrow \left[\begin{array}{cc} 1 - \mathbf{1}\boldsymbol{\alpha}^\top, & \boldsymbol{\alpha} \end{array}\right] \left[\begin{array}{c} \boldsymbol{g}_1 \\ \boldsymbol{G} \end{array}\right] \boldsymbol{U}^\top = \mathbf{0}, \tag{10}$$

where $\boldsymbol{U}^\top = \left[\boldsymbol{u}_1^\top - \boldsymbol{u}_2^\top, \cdots, \boldsymbol{u}_1^\top - \boldsymbol{u}_T^\top\right]$, $\mathbf{1}$ and $\mathbf{0}$ indicate the all-one and all-zero row vector, respectively. Eq. (10) can be solved by:

$$\left[\left(1 - \mathbf{1}\boldsymbol{\alpha}^\top\right)\boldsymbol{g}_1 + \boldsymbol{\alpha}\boldsymbol{G}\right]\boldsymbol{U}^\top = \mathbf{0} \Leftrightarrow \boldsymbol{\alpha}\left(\mathbf{1}^\top \boldsymbol{g}_1 - \boldsymbol{G}\right)\boldsymbol{U}^\top = \boldsymbol{g}_1 \boldsymbol{U}^\top. \tag{11}$$

Assume $\boldsymbol{D}^\top = \boldsymbol{g}_1^\top \mathbf{1} - \boldsymbol{G}^\top = \left[\boldsymbol{g}_1^\top - \boldsymbol{g}_2^\top, \cdots, \boldsymbol{g}_1^\top - \boldsymbol{g}_T^\top\right]$, then we reach:

$$\boldsymbol{\alpha}\boldsymbol{D}\boldsymbol{U}^\top = \boldsymbol{g}_1 \boldsymbol{U}^\top \Leftrightarrow \boldsymbol{\alpha} = \boldsymbol{g}_1 \boldsymbol{U}^\top \left(\boldsymbol{D}\boldsymbol{U}^\top\right)^{-1}. \tag{12}$$

**Property.** We can also prove the aggregated gradient $\boldsymbol{g} = \sum_t \alpha_t \boldsymbol{g}_t$ with $\{\alpha_t\}$ given in Eq. (12) is invariant to the scale changes of losses $\{L_t\}$ (or gradients $\{\boldsymbol{g}_t = \nabla_{\boldsymbol{\theta}} L_t\}$), as the following theorem.

**Theorem 2.** *Given* $\boldsymbol{g} = \sum_t \alpha_t \boldsymbol{g}_t$, $\sum_t \alpha_t = 1$ *satisfying* $\boldsymbol{g}\boldsymbol{u}_t^\top = C$, *when* $\{L_t\}$ *are scaled by* $\{k_t > 0\}$ *(equivalently,* $\{\boldsymbol{g}_t\}$ *are scaled by* $\{k_t\}$*), if* $\boldsymbol{g}' = \sum_t \alpha_t' \left(k_t \boldsymbol{g}_t\right)$, $\sum_t \alpha_t' = 1$ *satisfies* $\boldsymbol{g}'\boldsymbol{u}_t^\top = C'$, *then* $\boldsymbol{g}' = \lambda \boldsymbol{g}$. *In the above we have* $\boldsymbol{u}_t = \frac{\boldsymbol{g}_t}{\|\boldsymbol{g}_t\|} = \frac{k_t \boldsymbol{g}_t}{\|k_t \boldsymbol{g}_t\|}$, $\lambda$, $C$ *and* $C'$ *are constants.*

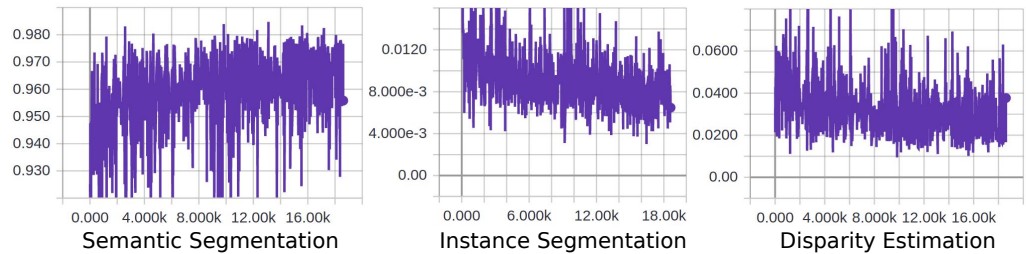

Figure 4: Loss scales of IMTL-G for different tasks when training on the Cityscapes dataset.

*Proof.* As we have:

$$\boldsymbol{g} = \sum_t \alpha_t \boldsymbol{g}_t = \sum_t \frac{\alpha_t}{k_t} k_t \boldsymbol{g}_t \quad \text{and} \quad \boldsymbol{g} \boldsymbol{u}_t^\top = C, \tag{13}$$

by constructing:

$$\alpha_t' = \frac{\alpha_t}{k_t} / \sum_\tau \frac{\alpha_\tau}{k_\tau} \quad \text{and} \quad \boldsymbol{g}' = \sum_t \alpha_t' (k_t \boldsymbol{g}_t) = \boldsymbol{g} / \sum_\tau \frac{\alpha_\tau}{k_\tau} = \lambda \boldsymbol{g}, \tag{14}$$

we have:

$$\sum_t \alpha_t' = 1 \quad \text{and} \quad \boldsymbol{g}' \boldsymbol{u}_t^\top = C / \sum_\tau \frac{\alpha_\tau}{k_\tau} = C'. \tag{15}$$

From Eq. (12) we know that $\{\alpha_t\}$ has a unique solution, and thus $\boldsymbol{g}'$ satisfying IMTL-G is unique, so it must be the one given by Eq. (14), then we can prove that $\boldsymbol{g}'$ and $\boldsymbol{g}$ are linearly correlated. □

### B.2 Loss balance: IMTL-L

With the ordinary differential equation, we can derive that the form of the scale function $\int h(s) \, \mathrm{d}s$ in our IMTL-L must be exponential function. As we have:

$$\int h(s) \, \mathrm{d}s = C h(s), \; C > 0. \tag{16}$$

If we set $y = \int h(s) \, \mathrm{d}s$, then:

$$y = C \frac{\mathrm{d}y}{\mathrm{d}s} \Rightarrow \frac{\mathrm{d}y}{y} = \frac{1}{C} \mathrm{d}s, \tag{17}$$

By taking the antiderivative:

$$\int \frac{\mathrm{d}y}{y} = \frac{1}{C} \int \mathrm{d}s \Rightarrow \ln y = \frac{1}{C} s + C'. \tag{18}$$

Then we have:

$$\int h(s) \, \mathrm{d}s = y = e^{C'} \left( e^{\frac{1}{C}} \right)^s = b a^s, \; a > 1, \; b > 0. \tag{19}$$

## C Detailed Discussion

### C.1 Conditional Equivalence of IMTL-G and GradNorm

First we introduce the following lemma.

**Lemma 3.** *If $\boldsymbol{u}_t \boldsymbol{u}_\tau^\top = C_1, \; \forall t \neq \tau$, then the solution $\{\alpha_t\}$ of IMTL-G satisfies $\{\alpha_t > 0\}$.*

*Proof.* As $\boldsymbol{u}_t = \boldsymbol{g}_t / \|\boldsymbol{g}_t\|$, by constructing $\boldsymbol{g} = \sum_t \alpha_t \boldsymbol{g}_t$ where:

$$\alpha_t = \|\boldsymbol{g}_t\|^{-1} / \sum_\tau \|\boldsymbol{g}_\tau\|^{-1}, \tag{20}$$

then we have $\sum_t \alpha_t = 1$ and:

$$\boldsymbol{g}\boldsymbol{u}_t^\top = \left(\sum_\tau \boldsymbol{u}_\tau \boldsymbol{u}_t\right) / \sum_\tau \|\boldsymbol{g}_\tau\|^{-1} = [(T-1)\,C_1 + 1] / \sum_\tau \|\boldsymbol{g}_\tau\|^{-1} = C_2. \tag{21}$$

From Eq. (12) we know the solution $\{\alpha_t\}$ of IMTL-G is unique, so it must be the one given by Eq. (20) where $\{\alpha_t > 0\}$, so the lemma is proved. $\qquad\square$

Then we prove Theorem 1 which states that IMTL-G leads to the same solution as GradNorm when the angle between any pair of gradients $\{\boldsymbol{g}_t\}$ is identical: $\boldsymbol{u}_t \boldsymbol{u}_\tau^\top = C_1, \ \forall t \neq \tau$.

*Proof.* ($\Rightarrow$ Necessity) Given constant projections in IMTL-G, we have:

$$\boldsymbol{g}\boldsymbol{u}_t^\top = \left(\sum_\tau \alpha_\tau \boldsymbol{g}_\tau\right) \boldsymbol{u}_t^\top = C_2. \tag{22}$$

Recall that $\boldsymbol{u}_t = \boldsymbol{g}_t / \|\boldsymbol{g}_t\|$ and $\boldsymbol{u}_t \boldsymbol{u}_\tau^\top = C_1, \ \forall t \neq \tau$. From Lemma 3 we know that $\{\alpha_t\}$ given by IMTL-G must satisfy $\{\alpha_t > 0\}$. If we assume $n_t = \|\alpha_t \boldsymbol{g}_t\|$, then we know $\alpha_t \boldsymbol{g}_t = n_t \boldsymbol{u}_t$ and:

$$\sum_\tau n_\tau \boldsymbol{u}_\tau \boldsymbol{u}_t^\top = \sum_{\tau \neq t} n_\tau C_1 + n_t = C_2. \tag{23}$$

Now we obtain:

$$\sum_{\tau \neq t} n_\tau C_1 + n_t = \sum_\tau n_\tau C_1 + (1 - C_1)\, n_t = C_2. \tag{24}$$

As $C_1 < 1$, we can then prove $n_t = C_3, \ \forall t$. It implies the norm of the scaled gradient is constant, which is requested by GradNorm (Chen et al., 2018). Moreover, we can obtain the relationship among constants from Eq. (24):

$$C_1 T C_3 + (1 - C_1)\, C_3 = C_2 \Rightarrow C_3 = \frac{C_2}{(T-1)\, C_1 + 1}. \tag{25}$$

($\Leftarrow$ Sufficiency) In GradNorm, $\{\alpha_t\}$ are always chosen to satisfy $\{\alpha_t > 0\}$, so if we assume $n_t = \|\alpha_t \boldsymbol{g}_t\|$, then given the constant norm of the scaled gradient in GradNorm, we have:

$$\alpha_t \boldsymbol{g}_t = n_t \boldsymbol{u}_t = C_3 \boldsymbol{u}_t, \tag{26}$$

where $\boldsymbol{u}_t = \boldsymbol{g}_t / \|\boldsymbol{g}_t\|$. As we have $\boldsymbol{g} = \sum_t \alpha_t \boldsymbol{g}_t$ and $\boldsymbol{u}_t \boldsymbol{u}_\tau^\top = C_1, \ \forall t \neq \tau$, then we obtain:

$$\boldsymbol{g}\boldsymbol{u}_t^\top = \left(\sum_\tau \alpha_\tau \boldsymbol{g}_\tau\right) \boldsymbol{u}_t^\top = \left(\sum_\tau C_3 \boldsymbol{u}_\tau\right) \boldsymbol{u}_t^\top = C_3 \left[(T-1)\, C_1 + 1\right] = C_2. \tag{27}$$

It means the projections of $\boldsymbol{g}$ onto $\{\boldsymbol{g}_t\}$ are constant, which is requested by our IMTL-G. $\qquad\square$

**Corollary 4.** *In GradNorm, if the solution $\{\alpha_t\}$ satisfies $\sum_t \alpha_t = 1$, then its constants are given by $C_3 = 1/\sum_t \|\boldsymbol{g}_t\|^{-1}$ and $C_2 = [(T-1)\, C_1 + 1]/\sum_t \|\boldsymbol{g}_t\|^{-1}$, and its scaling factors are given by $\left\{\alpha_t = \|\boldsymbol{g}_t\|^{-1} / \sum_\tau \|\boldsymbol{g}_\tau\|^{-1}\right\}$.*

*Proof.* By using $\alpha_t = C_3 / \|\boldsymbol{g}_t\|$ from Eq. (26), we have $\sum_t C_3 / \|\boldsymbol{g}_t\| = 1$, then $C_3 = 1/\sum_t \|\boldsymbol{g}_t\|^{-1}$, and also we have $\alpha_t = \|\boldsymbol{g}_t\|^{-1} / \sum_\tau \|\boldsymbol{g}_\tau\|^{-1}$. As the relationship of $C_2$ and $C_3$ from Eq. (27) is given by $C_3 [(T-1)\, C_1 + 1] = C_2$, so $C_2 = [(T-1)\, C_1 + 1] / \sum_t \|\boldsymbol{g}_t\|^{-1}$. $\quad\square$

## C.2    CLOSED-FORM SOLUTION OF MGDA

In our relaxed MGDA (Sener & Koltun, 2018) without $\{\alpha_t \geqslant 0\}$, finding $\boldsymbol{g} = \sum_t \alpha_t \boldsymbol{g}_t$ with $\sum_t \alpha_t = 1$ such that $\boldsymbol{g}$ has minimum norm is equivalent to find the normal vector of the hyperplane composed by $\{\boldsymbol{g}_t\}$. So we let $\boldsymbol{g}$ to be perpendicular to all of $\{\boldsymbol{g}_1 - \boldsymbol{g}_t\}$ on the hyper-plane:

$$\boldsymbol{g} \perp (\boldsymbol{g}_1 - \boldsymbol{g}_t) \Leftrightarrow \boldsymbol{g}(\boldsymbol{g}_1 - \boldsymbol{g}_t)^\top = 0, \ \forall \, 2 \leqslant t \leqslant T. \tag{28}$$

If we set $\boldsymbol{\alpha} = [\alpha_2, \cdots, \alpha_T]$ and $\boldsymbol{G}^\top = [\boldsymbol{g}_2^\top, \cdots, \boldsymbol{g}_T^\top]$, then we have $\alpha_1 = 1 - \boldsymbol{1}\boldsymbol{\alpha}^\top$, and Eq. (28) can be expanded as:

$$\left( \sum_t \alpha_t \boldsymbol{g}_t \right) \begin{bmatrix} \boldsymbol{g}_1^\top - \boldsymbol{g}_2^\top, & \cdots & , \boldsymbol{g}_1^\top - \boldsymbol{g}_T^\top \end{bmatrix} = \boldsymbol{0} \Leftrightarrow \begin{bmatrix} 1 - \boldsymbol{1}\boldsymbol{\alpha}^\top, & \boldsymbol{\alpha} \end{bmatrix} \begin{bmatrix} \boldsymbol{g}_1 \\ \boldsymbol{G} \end{bmatrix} \boldsymbol{D}^\top = \boldsymbol{0}, \tag{29}$$

where $\boldsymbol{D}^\top = [\boldsymbol{g}_1^\top - \boldsymbol{g}_2^\top, \cdots, \boldsymbol{g}_1^\top - \boldsymbol{g}_T^\top]$, $\boldsymbol{1}$ and $\boldsymbol{0}$ indicates the all-one and all-zero row vector. Eq. (29) can be represented as:

$$\left[ (1 - \boldsymbol{1}\boldsymbol{\alpha}^\top) \boldsymbol{g}_1 + \boldsymbol{\alpha}\boldsymbol{G} \right] \boldsymbol{D}^\top = \boldsymbol{0} \Leftrightarrow \boldsymbol{\alpha} \left( \boldsymbol{1}^\top \boldsymbol{g}_1 - \boldsymbol{G} \right) \boldsymbol{D}^\top = \boldsymbol{g}_1 \boldsymbol{D}^\top.$$

As we also have $\boldsymbol{D} = \boldsymbol{1}^\top \boldsymbol{g}_1 - \boldsymbol{G}$, then the closed-form solution of $\boldsymbol{\alpha}$ is given by:

$$\boldsymbol{\alpha}\boldsymbol{D}\boldsymbol{D}^\top = \boldsymbol{g}_1 \boldsymbol{D}^\top \Leftrightarrow \boldsymbol{\alpha} = \boldsymbol{g}_1 \boldsymbol{D}^\top \left( \boldsymbol{D}\boldsymbol{D}^\top \right)^{-1}. \tag{30}$$

**Bias of MGDA.** In the main text we state that MGDA focuses on tasks with small gradient magnitudes, where we relaxed MGDA by not constraining $\{\alpha_t \geqslant 0\}$. However, even with these constraints, the problem still exists. For example in the context of two tasks, assume $\|\boldsymbol{g}_1\| < \|\boldsymbol{g}_2\|$, if the minimum-norm point of $\boldsymbol{g}$ satisfying $\boldsymbol{g} = \alpha\boldsymbol{g}_1 + (1 - \alpha)\boldsymbol{g}_2$ is outside the convex hull composed by $\{\boldsymbol{g}_1, \boldsymbol{g}_2\}$, or equivalently $\alpha > 1$, MGDA clamps $\alpha$ to $\alpha = 1$ and the optimal $\boldsymbol{g}^\star = \boldsymbol{g}_1$. Then the projections of $\boldsymbol{g}^\star$ onto $\boldsymbol{g}_1$ and $\boldsymbol{g}_2$ will be $\|\boldsymbol{g}_1\|$ and $\boldsymbol{g}_1 \boldsymbol{u}_2^\top$ ($\boldsymbol{u}_2 = \boldsymbol{g}_2/\|\boldsymbol{g}_2\|$), respectively. As $\|\boldsymbol{g}_1\| > |\boldsymbol{g}_1 \boldsymbol{u}_2^\top|$, so MGDA still focuses on tasks with smaller gradient magnitudes.

## C.3    ANALYSIS OF PCGRAD

PCGrad (Yu et al., 2020) mitigates the gradient conflicts by projecting the gradient of one task to the orthogonal direction of the others, and the aggregated gradient can be written as:

$$\boldsymbol{g} = \sum_t \left( \boldsymbol{g}_t + \sum_\tau C_{t\tau} \boldsymbol{u}_\tau \right), \tag{31}$$

with $\boldsymbol{u}_t = \boldsymbol{g}_t/\|\boldsymbol{g}_t\|$ and the coefficients:

$$C_{tt} = 0, \ C_{t\tau} = \left[ -\left( \boldsymbol{g}_t + \sum_{t' < \tau,} C_{tt'} \boldsymbol{u}_{t'} \right) \boldsymbol{u}_\tau^\top \right]_+, \ \forall t, \tau, \tag{32}$$

where $[\cdot]_+$ means the ReLU operator. Note that the tasks have been shuffled before calculating the aggregated gradient $\boldsymbol{g}$ to achieve expected symmetry with respect to the task order. Eq. (31) can be represented more compactly in the matrix form:

$$\boldsymbol{g} = \boldsymbol{1}\left( \boldsymbol{I}_T + \boldsymbol{C}\boldsymbol{N} \right) \boldsymbol{G} \equiv \boldsymbol{\alpha}\boldsymbol{G}, \tag{33}$$

where $\boldsymbol{I}_T$ is the identity matrix, $\boldsymbol{C} = \{C_{t\tau}\}$ is the coefficient matrix whose entries are given in Eq. (32) and $\boldsymbol{N} = \mathrm{diag}\left(1/\|\boldsymbol{g}_1\|, \cdots, 1/\|\boldsymbol{g}_T\|\right)$ is the diagonal normalization matrix. In Eq. (33) we use $\boldsymbol{G}$ and $\boldsymbol{\alpha}$ to denote the raw gradients and scaling factors of all tasks. We find that PCGrad can also be regarded as loss weighting, with the loss weights given by $\boldsymbol{\alpha} = \boldsymbol{1}\left( \boldsymbol{I}_T + \boldsymbol{C}\boldsymbol{N} \right)$. However, it still may break the balance among tasks. For example with two tasks, assume the angle between

the gradients is $\phi$: 1) when $\pi/2 \leqslant \phi < \pi$, then $\boldsymbol{C} = \begin{bmatrix} 0 & -\boldsymbol{g}_1\boldsymbol{g}_2^\top / \|\boldsymbol{g}_2\| \\ -\boldsymbol{g}_1\boldsymbol{g}_2^\top / \|\boldsymbol{g}_1\| & 0 \end{bmatrix}$ and the projections onto the two raw gradients are $\|\boldsymbol{g}_1\| \sin^2 \phi$ and $\|\boldsymbol{g}_2\| \sin^2 \phi$; 2) when $0 < \phi < \pi/2$, then $\boldsymbol{C} = \boldsymbol{0}$ and the projections are $\|\boldsymbol{g}_1\| + \|\boldsymbol{g}_2\| \cos \phi$ and $\|\boldsymbol{g}_2\| + \|\boldsymbol{g}_1\| \cos \phi$. In both cases, the projections are equal if and only if $\|\boldsymbol{g}_1\| = \|\boldsymbol{g}_2\|$. Otherwise, the task with larger gradient magnitude will be trained more sufficiently, which may encounter the same problem as uniform scaling that naïvely adds all the losses despite that the loss scales are highly different.

### C.4 $L_2$ LOSS IN UNCERTAINTY WEIGHTING

For regression, uncertainty weighting (Kendall et al., 2018) regards the $L_2$ loss as likelihood estimation on the sample target which follows the Gaussian distribution:

$$- \log p \left( y \mid f \left( \boldsymbol{x} \right) \right) = \frac{1}{2} \left( \frac{1}{\sigma^2} \|y - f \left( \boldsymbol{x} \right)\|_2^2 + \log \sigma^2 \right), \tag{34}$$

where $\boldsymbol{x}$ is the data sample, $y$ is the ground-truth label, $f$ denotes the prediction model and $\sigma$ is the standard deviation of Gaussian distribution. By setting $s = -\log \sigma^2$, the scaled $L_2$ loss is $L = \frac{1}{2} \left( e^s L_{\text{reg}} - s \right)$, which has a similar form as the scaled $L_1$ loss except the front factor $1/2$. So uncertainty weighting has difficulty in reaching a unified form for all kinds of losses, which is less general than our IMTL-L.

### C.5 GRADIENT OF GEOMETRIC MEAN

GLS (Chennupati et al., 2019) computes the loss as the geometric mean, its gradient with respect to model parameters are:

$$\nabla_{\boldsymbol{\theta}} L = \frac{1}{T} \left( \prod_t L_t \right)^{\frac{1}{T} - 1} \sum_t \left[ \left( \prod_{\tau \neq t} L_\tau \right) \nabla_{\boldsymbol{\theta}} L_t \right] \tag{35}$$

$$= \frac{1}{T} \left( \prod_t L_t \right)^{\frac{1}{T}} \sum_t \frac{\nabla_{\boldsymbol{\theta}} L_t}{L_t} = \frac{L}{T} \sum_t \frac{1}{L_t} \left( \nabla_{\boldsymbol{\theta}} L_t \right). \tag{36}$$

where $L$ is the geometric mean loss and $T$ is the task number. It is equivalent to weigh the task-specific loss with its reciprocal value, except that there exists another term $L/T$ in the front where $L = \left( \prod_t L_t \right)^{\frac{1}{T}}$, so GLS is sensitive to the loss scale changes of $\{L_t\}$ and not scale-invariant.

## D IMPLEMENTATION DETAILS

To solely compare the loss weighting methods, we fix the network structure and choose ResNet-50 (He et al., 2016) with dilation (Chen et al., 2017) and synchronized (Peng et al., 2018) batch normalization (Ioffe & Szegedy, 2015) as the shared backbone and PSPNet (Zhao et al., 2017) as the task-specific head, and the backbone model weights are pretrained on ImageNet (Deng et al., 2009). Following the common practice of semantic segmentation, in training we adopt augmentations as random resize (between 0.5 to 2), random rotate (between -10 to 10 degrees), Gaussian blur (with a radius of 5) and random horizontal flip. Besides, we apply strided cropping and horizontal flipping as testing augmentations. The predicted results in the overlapped region of different crops are averaged to obtain the aggregated prediction of the whole image. Only pixels with ground truth labels are included in loss and metric computation, while others are ignored. Semantic segmentation, instance segmentation, surface normal estimation and disparity/depth estimation are considered. As for the losses/metrics, semantic segmentation uses cross-entropy/mIoU, surface normal estimation adopts $(1 - \cos)$/cosine similarity and both instance segmentation and disparity/depth estimation use $L_1$ loss. We use polynomial learning rate with a power of 0.9, SGD with a momentum of 0.9 and weight decay of $10^{-4}$ as the optimizer, with the model trained for 200 epochs. After passing through the shared backbone where strided convolutions exist, the feature maps have $1/8$ size as that of the

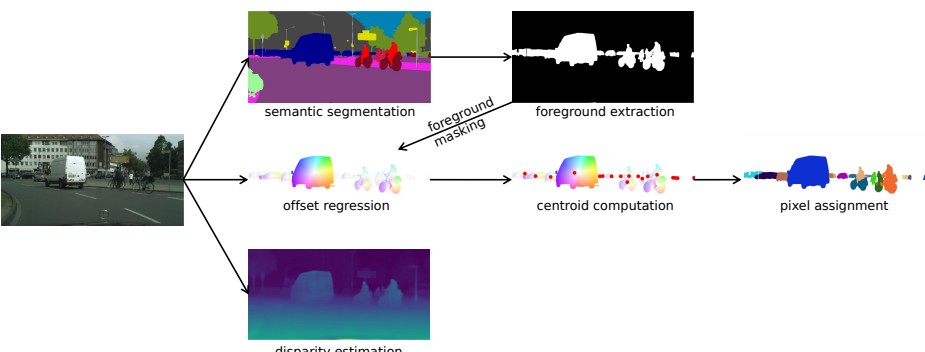

Figure 5: Pipeline used in the Cityscapes visual understanding experiment. The centroids are computed from the offset regression results. Each pixel is assigned to its nearest candidate centroid.

input image. Then the results predicted by PSPNet (Zhao et al., 2017) heads are up-sampled to the original image size for loss and metric computation.

For the **Cityscapes** dataset, the batch size is 32 ($2 \times 16$ GPUs) with the initial learning rate 0.02. We train on the 2975 training images and validate on the 500 validation images ($1024 \times 2048$ full resolution) where ground truth labels are provided. Three tasks are considered, namely semantic segmentation, instance segmentation and disparity/depth estimation. Training and testing are done on $713 \times 713$ crops. Semantic segmentation is to differentiate among the commonly used 19 classes. Instance segmentation is taken as offset regression, where each pixel $\boldsymbol{p}_i = (x_i, y_i)$ approximates the relative offset $\boldsymbol{o}_i = (\mathrm{d}x_i, \mathrm{d}y_i)$ with respect to the centroid $\boldsymbol{c}_{\mathrm{id}(\boldsymbol{p}_i)}$ of its belonging instance $\mathrm{id}(\boldsymbol{p}_i)$. To conduct inference, we abandon the time-consuming and complicated clustering methods adopted by the previous method (Kendall et al., 2018). Instead, we directly use the offset vectors $\{\boldsymbol{o}_i\}$ predicted by the model to find the centroids of instances. By definition, the norm of a centroid's offset vector should be 0, so we can transform the offset vector norm $\|\boldsymbol{o}_i\|$ to the probability $q_i$ of being a centroid with the exponential function $q_i = e^{-\|\boldsymbol{o}_i\|}$. Next a $7 \times 7$ edge filter is applied on the centroid probability map to filter out the spurious centroids on object edges resulting from the regression target ambiguity. The locations with centroid probability $q_i < 0.1$ are also manually suppressed. Then $7 \times 7$ max-pooling on the filtered probability map is used to produce candidate centroids and filter out duplicate ones. With the predicted centroids $\{\boldsymbol{c}_i\}$, we can then assign each pixel $\boldsymbol{p}_i$ to its belonging instance $\mathrm{id}(\boldsymbol{p}_i)$ by the distance between its approximated centroids $\boldsymbol{p}_i + \boldsymbol{o}_i$ and the candidate centroids $\{\boldsymbol{c}_i\}$: $\mathrm{id}(\boldsymbol{p}_i) = \arg\min_j \|\boldsymbol{p}_i + \boldsymbol{o}_i - \boldsymbol{c}_j\|$. Depth is measured in pixels by the disparity between the left and right images. Fig. 5 shows the whole process. Note that we need to carefully deal with label transformation during data augmentation. For example, disparity ground truth needs to be up-scaled by $s$ times if the image is up-sampled by $s$ times. Also, the predicted offset vectors of the flipped input should be mirrored to comply with the normal one.

On the **NYUv2** dataset, the batch size is 48 ($6 \times 8$ GPUs) with the initial learning rate 0.03. We use the 795 training images for training and the 654 validation images for testing with $480 \times 640$ full resolution. $401 \times 401$ crops are used for training and testing. 13 coarse-grain classes are considered in semantic segmentation. The surface normal is represented by the unit normal vector of the corresponding surface. When doing data augmentation, surface normal ground truth $\boldsymbol{n} = (x, y, z)$ should be processed accordingly. If we resize the image by $s$ times, the $z$ coordinate of the normal vector should be scaled by $s$ and renormalized: $\boldsymbol{n}' = (x, y, sz) / \|(x, y, sz)\|$. If the image is rotated by the rotation matrix $\boldsymbol{R}$, the normal vector should also be in-plane rotated $(x', y') = (x, y) \boldsymbol{R}^{\top}$ with $z$ unchanged. Moreover, the left-right flip should be applied on the normal vector $\boldsymbol{n}' = (-x, y, z)$ when mirroring the image horizontally. During testing, the normal vectors in the overlapped region of crops are averaged and renormalized to produce the aggregated results. Depth is the absolute distance to the camera and measured by meters, which is inverse-proportional to the disparity measurement adopted by Cityscapes. So the depth in meters needs to be scaled by $1/s$ when the image is scaled by $s$ times, which is the reciprocal of disparity transformation.

**CelebA** contains 202,599 face images from 10,177 identities, where each image has 40 binary attribute annotations. We train on the 162,770 training images and test on the 19,867 validation

images. Most of the implementation details are the same as those on the Cityscapes dataset, except that: 1) we employ the ResNet-18 as the backbone and linear classifiers as the task-specific heads, so totally 40 heads are attached on the backbone ; 2) the binary-cross entropy is used as the classification loss for each attribute; 3) the batch size is 256 ($32 \times 8$ GPUs) and the model is trained from scratch for 100 epochs; 4) the input image has been aligned with the annotated 5 landmarks and cropped to $218 \times 178$.

# E  QUALITATIVE RESULTS

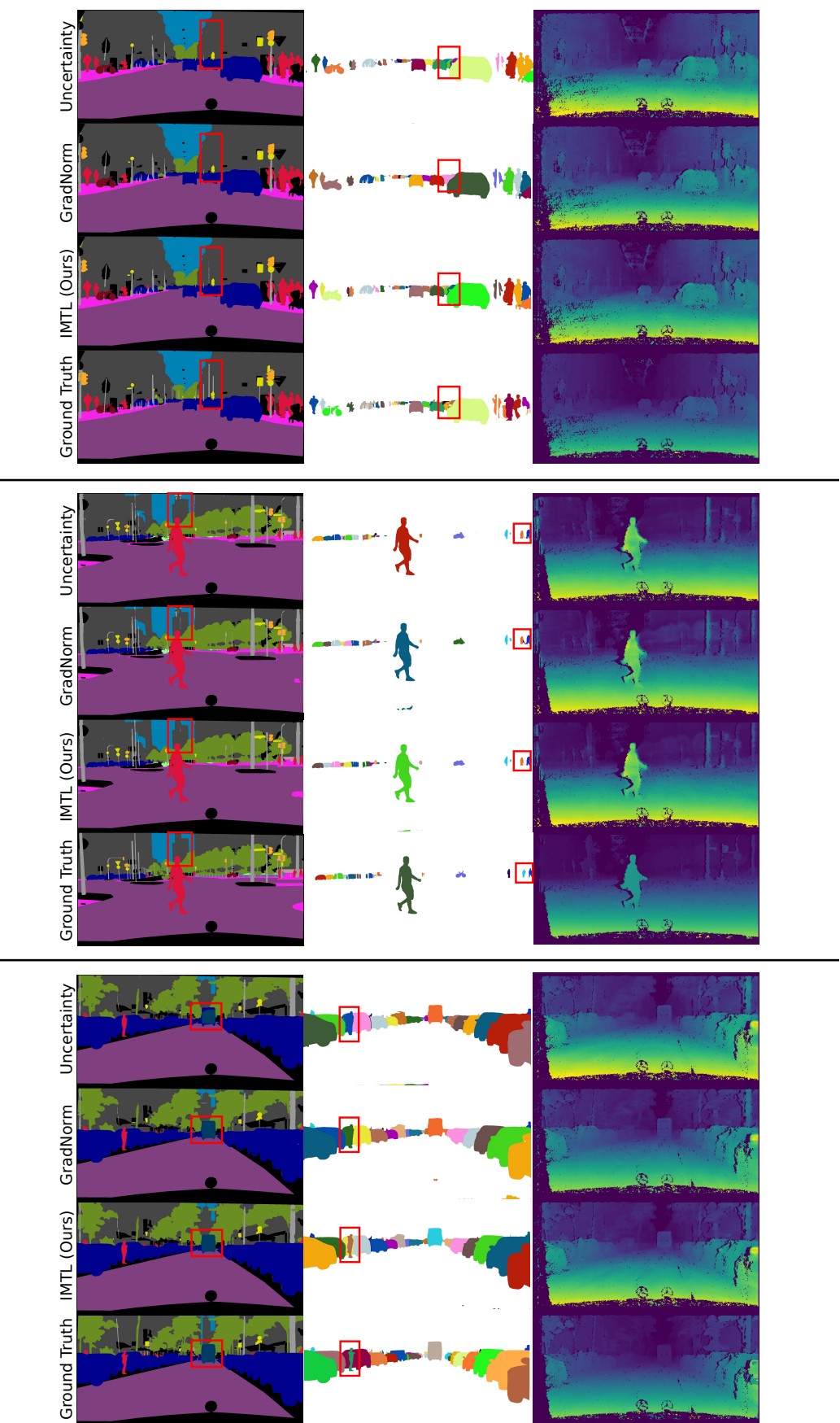

Figure 6: Qualitative comparisons between our IMTL and previous methods on Cityscapes.

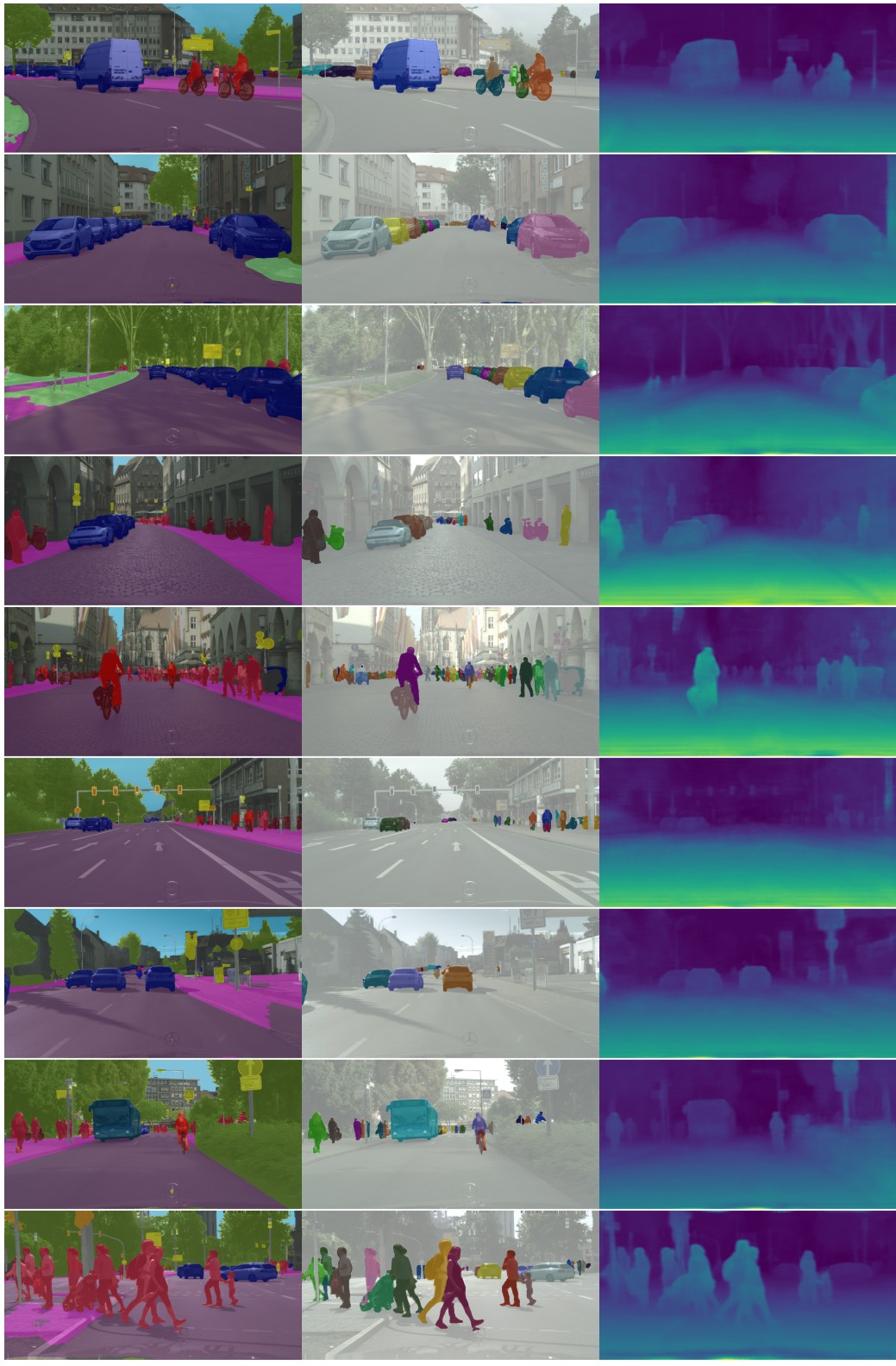

Figure 7: Qualitative results of our IMTL on Cityscapes. Semantic segmentation, instance segmentation and disparity estimation predictions are produced by a single network. The task-shared backbone is ResNet-50 and the task-specific heads are PSPNet. The image resolution is $1024 \times 2048$.

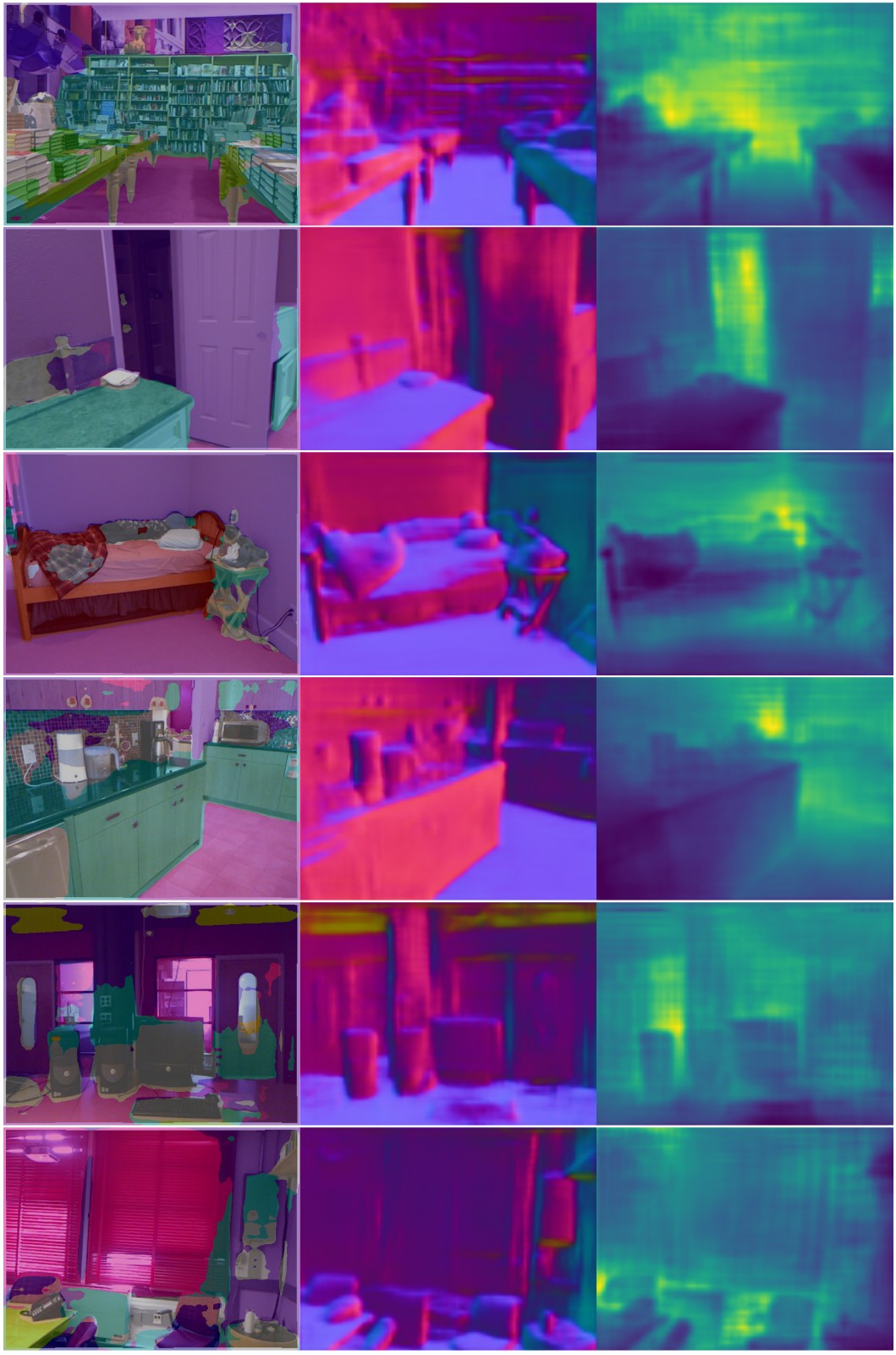

Figure 8: Qualitative results of our IMTL on NYUv2. Semantic segmentation, surface normal estimation and depth estimation predictions are produced by a single network. The task-shared backbone is ResNet-50 and the task-specific heads are PSPNet. The image resolution is $480 \times 640$.

