# OpenReview forum: "Towards Impartial Multi-task Learning"
_ICLR.cc/2021/Conference — ICLR 2021 Poster_

### Official Review · AnonReviewer3 · 2020-10-19
**Interesting idea but lacks correctness**

**Rating:** 4
**Confidence:** 5

**Review:**

The authors propose to balance multi-task training using IMTL-G on the shared backbone and IMTL-L on the task-specific branches. IMTL-G enforces equal gradient projections between tasks with a close-form formulation to calculate the desired gradient weightings $\alpha$. IMTL-L learns the loss weightings $e^s$ with a regularization term $-s$. Additional constraint by making all loss weightings sum to one is used. The paper compares the effectiveness of the proposed IMTLs with their counterparts on Cityscapes, NYUv2, and CelebA and claims state-of-the-art performance.

First of all, the authors start out by claiming multi-task learning architecture improvement will lead to high inference cost (Sec 2 first paragraph), so they decided to investigate loss weighting balancing. However, methods have been developed to explore better task hierarchies to design better task grouping in multi-task network architectures in recent publications [R1, R2]. They show SOTA results without high inference cost. In fact, these works also show that better task grouping in network design can already alleviate the need to use sophisticated loss balancing tricks. Although multi-task network design and loss balancing can be orthogonal efforts, it is important to have reasonable motivation and fair discussion.

The authors claim state-of-the-art performance on all tasks. However, the proposed IMTL method obtained 91.12% accuracy on CelebA where method in [R2] obtained 91.62% accuracy. Therefore, it is incorrect to claim SOTA in the submission.

Equation (2) requires expensive matrix inverse operation for every iteration during training. How much additional computational overhead is needed compared to other straightforward gradient balancing approaches? It would be great to have a clear time complexity comparison with real world runtimes measured.

The authors claim the proposed IMTL-L does not require any distribution assumption, resulting in the obtained loss weighting $e^s$ with a regularization term $-s$. The proposed formulation is actually quite similar to the original uncertainty weighting approach (Kendall et al. v1 eq.11) with the Gaussian distribution assumption. The only difference is the 0.5 scale for the first term. This similarity also reflected in the experimental results that in Table 1 we can see IMTL-L performs quite similarly to uncertainty weighting (Kendall et al). Uncertainty weighting has even better performance on instance segmentation and disparity compared to the IMTL-L. Can the authors discuss in more details the differences between the two approaches?

Besides, I found Algorithm 1 on page 4 is quite straightforward. It would be great to use the space to further analysis the difference between the proposed IMTL-L and the uncertainty weighting approach (Kendall et al).

[R1] Standley et al. Which Tasks Should Be Learned Together in Multi-task Learning? In ICML 2020. \
[R2] Guo et al. Learning to Branch for Multi-Task Learning. In ICML 2020.

---

> ### Author Response · Authors · 2020-11-21
> **To AnonReviewer3**
>
> Q1: Fair discussion of methods that improve network structures.
> A1: Thank you for your review and comments. We have revised in Sec. 2 as "There are also efforts that learn to group tasks and branch the network in the middle layers (Guo et al., 2020; Standley et al., 2020), which try to achieve better accuracy-efficiency trade-off and can be seen as semi-hard parameter-sharing. We believe task grouping and loss weighting are orthogonal and complementary directions to facilitate multi-task learning and can benefit from each other." We would also like to stress that although task grouping methods share a greater number of parameters/computations than soft parameter sharing, each task has its own parameters in the backbone. Hard parameter-sharing based on loss weighting is still the most efficient one in terms of inference speed as even more (almost all of) parameters/computations are shared among tasks.
>
> Q2: Claim of the SoTA results.
> A2: Thank you for pointing this out. We claim state-of-the-art results in the context of loss weighting under fair settings, and we have revised our statement in the abstract as "It achieves the new state-of-the-art **among loss weighting methods** under the same experimental settings." We believe methods which improve network structures can bring good results and they may benefit from our IMTL to further boost the performance. However, it may not be fair to directly compare results of methods that improve network structure and loss weighting as the experimental settings like image resolution, backbone architecture, optimizer option and learning rate scheduler may be different. In our work we compare with previous methods under the same setting to ensure fairness and verify our superiority.
>
> Q3: Time complexity comparisons.
> A3: We appreciate your suggestion. "We present the real-world training time of each iteration for different methods in Tab. 1. As shown, loss balance methods are the most efficient, and our gradient balance method IMTL-G adds acceptable computational overhead, similar to that of GradNorm (Chen et al., 2018) and MGDA (Sener and Koltun, 2018). It benefits from computing gradients with respect to the shared feature maps instead of the shared model parameters." (revision in Sec. 5) Note that in Eq. (2) the computation overhead mainly comes from the matrix multiplication rather than the matrix inverse, since the inversed matrix $\boldsymbol{D}\boldsymbol{U}^{\top}\in\mathbb{R}^{\left(T-1\right)\times\left(T-1\right)}$ is small. The multiplication has also been accelerated by the approximated gradients on shared feature maps rather than shared parameters.
>
> Q4: Differences with uncertainty weighting.
> A4: The suggested comparison between IMTL-L and uncertainty weighing (Kendall et al., 2018) is added in Sec. 4. "**Firstly**, our derivation is motivated by the fairness among tasks, which intrinsically differs from uncertainty weighting which is based on task uncertainty considering each task independently. **Secondly**, IMTL-L learns to balance among tasks without any biases, while uncertainty weighting may sacrifice classification tasks to favor regression tasks as derived above. **Thirdly**, IMTL-L does not depend on any distribution assumptions and thus can be generally applied to various losses including cosine similarity, which uncertainty weighting may have difficulty with. As far as we know, there is no appropriate correspondence between cosine similarity and specific distributions. **Lastly**, uncertainty weighting needs to deal with different losses case by case, it also introduces approximations in order to derive scaling factors for certain losses (such as cross-entropy loss) which may not be optimal, but our IMTL-L has a unified form for all kinds of losses."
>
> Q5: Performances of uncertainty weighting.
> A5: As derived, in uncertainty weighting the classification task (semantic segmentation) is down-weighted and can act as an auxiliary task, so it's reasonable that it performs **slightly** better for regression tasks (instance segmentation and disparity estimation) via sacrificing the classification task due to the limited model capacity. On the other hand, we outperform uncertainty weighting on semantic segmentation **significantly**. Our method achieves the best trade-off among all tasks without bias towards any specific one, so that each task shows impressive results.

---

### Official Review · AnonReviewer4 · 2020-10-23
**Interesting work but not that significant**

**Rating:** 5
**Confidence:** 5

**Review:**

This paper introduces an impartial multi-task learning approach to balance gradient and loss of multiple tasks. Since biased learning can degrade learning efficiency and this work tries to make a balance in a principled way. The approach is simple but looks effective.
However, the gradient balance adopts the last shared feature instead of weights for faster training. From this, I wonder whether this kind of approximation decreases the capability of the weight-level balancing strategy. Theoretical or empirical analyses on this make the proposed method stronger.

The proposed method gives comparable performance compared to other strong competitors but the gap is not remarkable. I wonder if this performance gap can deserve the attention of practitioners. It would be great for more rigorous analyses to show qualitative results and how this approach differentiates it from others. The proposed method needs to show its potential further.

In addition, if this approach applies to multiple datasets (as multiple tasks, for example, visual decathlon challenge), the proposed approach may not work well due to the different characteristics when we train them using a hard parameter sharing approach. For a general approach, it would be great to analyze a variety of scenarios.

---

> ### Author Response · Authors · 2020-11-21
> **To AnonReviewer4**
>
> Q1: Effectiveness of the gradient approximation.
> A1: Due to the chain rule of gradient computation, we believe the approximation is effective and will not limit the capability of our gradient balance method. In Tab. 1 we add the results of computing gradients with respect to shared model parameters $\boldsymbol{\theta}$ (the row of “IMTL-G (exact)”) instead of share feature maps $\boldsymbol{z}$, it brings similar performances as our approximation but adds significant complexity due to multiple (T) backward passes through the shared parameters. Our IMTL-G only needs to do backward computation on the shared parameters once after obtaining the loss weights. We find that the results obtained by approximated gradients {$\nabla_{\boldsymbol{z}}L_{t}$}  are even better than those of {$\nabla_{\boldsymbol{\theta}}L_{t}$}, it may result from the use of SGD where we compute the gradients with a batch of samples instead of the whole dataset. If the computed gradient introduces a **random** error $\widehat{\nabla_{\boldsymbol{\theta}}L_{t}}=\nabla_{\boldsymbol{\theta}}L_{t}+e_t$, from the stability analysis (Sener and Koltun, 2018) we know the error in the loss weighting coefficients is bounded $\lVert\widehat{\boldsymbol{\alpha}}-\boldsymbol{\alpha}\rVert_2$$\leqslant\mathcal{O}$($\max_t\lVert e_t\rVert_2$). We can expect $\lVert e_t\rVert_2$ to depend on the dimension of $e_t$, and thus we may receive a more accurate solution via replacing $\nabla_{\boldsymbol{\theta}}L_{t}$ with $\nabla_{\boldsymbol{z}}L_{t}$ due to its smaller dimension. (revision in Sec. 5) The same conclusion can be found in (Sener and Koltun, 2018).
>
> Q2: Qualitative comparisons with previous methods.
> A2: Thank you for your suggestions. "As we outperform MGDA (Sener and Koltun, 2018) and PCGrad (Yu et al., 2020) significantly in terms of the objective metrics shown in Tab. 1, we further compare the qualitative results of our hybrid balance IMTL with the loss balance method uncertainty weighting (Kendall et al., 2018) and the gradient balance method GradNorm (Chen et al., 2018) considering their strong performances (see Fig. 5 in Appendix E). Note that for depth estimation we only show predictions at the pixels where ground truth (GT) labels exist to compare with GT, which is different from Fig. 6 where depth predictions are shown for all pixels. Consistent with results in Tab. 1, our IMTL shows visually noticeable improvements especially for the semantic and instance segmentation tasks. It is worth noting that we conduct experiments under strong baselines and practical settings which are seldom explored before, in this case changing the backbone in PSPNet (Zhao et al., 2017) from ResNet-50 to ResNet-101 can only improve mIoU of the semantic segmentation task around 0.5% on Cityscapes according to the public code base." (https://github.com/open-mmlab/mmsegmentation/tree/master/configs/pspnet) (revision in Sec. 5) We believe our method deserves attentions from practitioners.
>
>
> Q3: The multi-dataset multi-task setting.
> A3: We agree that multi-dataset multi-task learning is a valuable and general problem which is as important as multi-label multi-task learning considered in this work. We want to remark that we focus on the latter one and all the compared methods do not explore the multi-dataset setting. Extensive experments have been conducted, and for future work we would like to study the mentioned setting and see whether the hard parameter-sharing paradigm can perform well in this case. It is a possible direction to simultaneously learn task grouping and task balance to achieve the best trade-off between accuracy and efficiency. (revision in Sec. 6)

---

### Official Review · AnonReviewer2 · 2020-10-28
**A principled and practical solution to training multiple task in a fair way**

**Rating:** 7
**Confidence:** 4

**Review:**

Summary:

This paper presents a satisfying solution to the open problem of how to train all tasks at approximately the same rate in multi-task learning. There has been a bunch of work on this problem in the last few years. This paper characterizes existing work w.r.t. the fairness of training across tasks in order to motivate two new methods, one applied to shared parameters and the other to task-specific parameters, which overcome the shortcomings of previous methods. The two new methods can be naturally combined to yield a complete method for fair training. Experiments on common MTL benchmarks show the new method compares quite favorably to previous approaches.

The methods are developed in a theoretical manner from a reasonable set of assumptions. Despite the theoretical derivation, the resulting methods are straightforward to implement, especially compared to more complex iterative or probabilistic methods. Due to its high performance and simplicity of implementation, my expectation is that this new method could quickly become a standard tool for multi-task learning applications.


Strong points:

The paper is overall well-written and well-motivated in the literature.

The paper addresses an important open problem in multi-task learning.

The theoretical derivations are well-motivated based on fairness desiderata.

The resulting methods are effective and simple-to-implement.

The theoretical and conceptual comparisons to existing methods make it clear how the paper unifies and extends the existing literature.

The experiments are convincing, and the value of each aspect of the method is independently confirmed.


Weak points:

The paper could benefit from a discussion of the limitations of using fairness as a proxy for MTL performance. Fairness is an intuitive requirement, but one can imagine cases where some tasks should be trained more than others, e.g., if some tasks begin to overfit at higher losses than others.

The motivation is framed as addressing the problem of some tasks being sacrificed/undertrained. However, there is a common and closely-related issue of some tasks overfitting before others are satisfactorily trained. So, the more general question is how to synchronize the state of training so that all tasks reach peak performance at the same time. Does the approach in the paper address this more general question? Or is it only effective in the setting where overfitting is not an issue because the datasets are so large?

For example, in the paper the comment “NYUv2 is a rather small dataset, so uniform scaling can also obtain reasonable results” emphasizes that underfitting is the focus, but if the dataset is so small, could uniform scaling lead to overfitting that IMTL could address?

In the experiments, are all the key details of the existing methods reimplemented, or are some of them taken from official implementations? This should be made clear in the paper and in the released code.


Minor comments:

In the first paragraph of the introduction, are Zamir et al. 2018 and 2020 really the key references for MTL overall? There is no evidence from the introduction that MTL was around before 2018.

In the first full sentence after Eq. 8: “corresponds to maximize the” -> “corresponds to maximizing the”.

In “Results on Cityscapes”: “Surprisingly, we find our IMTL can beat the single-task baseline where each task is trained on a separate model.” I don’t think this is surprising. By your definition of MTL in the introduction, this is what you expect if you do MTL right. However, it is somewhat surprising that none of the other MTL methods exceed the single-task baseline on all three tasks; this highlights the fact that they are missing something.

In Appendix D: “The output size is initially 8x down-sampled and later up-sampled…” Why?

---------------------

Update: The authors have adequately addressed my questions, and I am happy to maintain the rating of 7.

---

> ### Author Response · Authors · 2020-11-21
> **To AnonReviewer2**
>
> Q1: The over-fitting problem.
> A1: Thank you for your encouraging comments and constructive advice. In our experiments we do not encounter the over-fitting issue with real-world datasets, and find under-fitting is the main reason that prevents the model from performing well on specific tasks. To the best of our knowledge, the over-fitting problem has not been reported in previous multi-task learning literature (Kendall et al., 2018; Chen et al., 2018; Sener and Koltun, 2018; Yu et al., 2020). The mentioned general problem of synchronizing training states among different tasks where both under-fitting and over-fitting may exist is indeed a practical and valuable direction. We would like to explore multi-task learning for more scenarios and compare different methods in our future work (revision in Sec. 6).
>
> Q2: Limitations of fairness.
> A2: As for limitations, we add the discussion in Sec. 4 as "Taking fairness among tasks as a proxy is not always beneficial, especially when one of the tasks is more important than others. In this case we may deliberately down-weight certain tasks to better aid the most concerned one." In this work, we follow previous loss weighting methods and focus on the setting where all tasks are equally important, and thus we adopt fairness as a reasonable proxy.
>
> Q3: Implementation of previous methods.
> A3: Thank you for your suggestion. "The implementations exactly follow the original papers and open-sourced code to ensure the correctness." (revision in Sec. 5) All the key details are **fully** re-implemented and will be made publicly available.
>
> Q4: General references for multi-task learning.
> A4: We appreciate your careful review. We cite (Zamir et al. 2018; Zamir et al. 2020) because they systematically study large-scale multi-task learning and show impressive results. We have also added earlier references such as (Caruana, 1997; Evgeniou & Pontil,
> 2004; Ruder, 2017; Zhang & Yang, 2017) in our revised manuscript (revision in Sec. 1). We have carefully proofread our submission and corrected the typos.
>
> Q5: Expectation of multi-task learning over single-task baselines.
> A5: We do expect multi-task learning to show improvements over single-task baselines. However, through experiments we find that this can **not always** be achieved especially under strong single-task baselines, or when methods have biases towards specific tasks. This phenomenon can also be found in (Standley et al., 2020). In this work we adopt practical settings which lead to strong single-task performances and are seldom explored before, the superior results of our method show the benefits of impartial learning. Although the compared methods can not surpass all single-task baselines, they show impressive results on certain tasks, verifying that our implementations are correct.
>
> Q6: Details of PSPNet for scene understanding.
> A6: The mentioned 8x down-sampling is because we adopt PSPNet (Zhao et al., 2017) for scene understanding. "After passing through the shared backbone where **strided convolutions** exist, the feature maps have 1/8 size as that of the input image. Then the results predicted by PSPNet heads are up-sampled to the original image size for loss and metric computation." (revision in Appendix D)

---

### Author Response · Authors · 2020-11-21
**To all reviewers**

We sincerely thank all the reviewers for your comments and suggestions, we have uploaded our revised manuscript where the revisions are in **blue**. Next we will try to answer the concerns for individual reviewers and hope our responses can meet your approval. If you have further questions, please comment below without hesitation.

---

### Decision · Program_Chairs · 2021-01-07
**Final Decision**

**Decision:**

Accept (Poster)

**Comment:**

The paper is proposing a multi-task learning approach extending existing weighting approaches. An important and novel contribution of the paper is separating the magnitude and direction information in gradient based information. The joint gradient direction is searched by using angle bisectors of task gradients and magnitude is searched by simply finding a scaling which results in uniform loss scales. This approach solves issues like small gradient norm bias of MGDA, etc. The proposed method works well and authors show that it is conceptually relevant to most of the existing algorithms. These conceptual unification is a strong contribution of the paper. The paper is reviewed by three reviewers and received both accept and reject scores. Specifically,

- R#2: Championed the paper and argued for its acceptance
- R#3: Argues that the novelty is limited and SOTA claim is problematic.
- R#4: Argues that the gap between the empirical performance of the proposed method and existing algorithms is small.

Arguments on the empirical performance and the SOTA are irrelevant to the decision since ICLR does not require algorithms to be SOTA or performed significantly better. Hence, the remaining issues are: claim of the SOTA being true or not, and lack of novelty. I read the paper in detail and decided to accept it with the following comments about the reviews:
- The paper is clearly novel. Direction and magnitude are first time treated separately. Moreover, resulting unification of the existing approaches and theoretical derivations of the important connections of existing methods are also significant.
- The SOTA claim of the paper is technically correct but little misleading. I would recommend authors to simply rephrase it "proposed method outperforms existing methods loss weighting methods under the same experimental settings". The reason for this is the fact that; in principle, "art" includes every possible solution for that problem. Hence, claiming SOTA in a fair and limited evaluation is rather misleading.

In addition to the reviewer comments, here are additional issues which should be addressed by the camera-ready deadline:
- I think the discussion about MGDA is a bit problematic since removing $\alpha \geq 0$ assumption simply removes the Pareto stationarity guarantee of the method. The resulting direction can increase some loss function and this disagrees with the main point of the Pareto optimality. Hence, I would recommend authors to clarify this while making the connection. Frank-Wolfe algorithm is also not really inefficient and unstable since the problem is quadratic with linear constraints and the stability as well as extremely quick convergence can trivially be proved.
- In addition to the previous point, the proposed method can actually increase some loss functions as there is no consistency constraint enforced. This is an interesting observation and empirical results suggest that increasing loss of some objectives might actually be valuable. I think this observation deserves some discussion even in the introduction.